# 3DTrajMaster: Mastering 3D Trajectory for Multi-Entity Motion in Video Generation

**Xiao Fu**[1†]  **Xian Liu**[1]  **Xintao Wang**[2✉]  **Sida Peng**[3]  **Menghan Xia**[2]  **Xiaoyu Shi**[2]
**Ziyang Yuan**[2]  **Pengfei Wan**[2]  **Di Zhang**[2]  **Dahua Lin**[1✉]
[1]The Chinese University of Hong Kong   [2]Kuaishou Technology   [3]Zhejiang University

## Abstract

This paper aims to manipulate multi-entity 3D motions in video generation. Previous methods on controllable video generation primarily leverage 2D control signals to manipulate object motions and have achieved remarkable synthesis results. However, 2D control signals are inherently limited in expressing the 3D nature of object motions. To overcome this problem, we introduce **3DTrajMaster**, a robust controller that regulates multi-entity dynamics in *3D space*, given user-desired 6DoF pose (location and rotation) sequences of entities. At the core of our approach is a plug-and-play 3D-motion grounded object injector that fuses multiple input entities with their respective 3D trajectories through a gated self-attention mechanism. In addition, we exploit an injector architecture to preserve the video diffusion prior, which is crucial for generalization ability. To mitigate video quality degradation, we introduce a domain adaptor during training and employ an annealed sampling strategy during inference. To address the lack of suitable training data, we construct a 360°-Motion Dataset, which first correlates collected 3D human and animal assets with GPT-generated trajectory and then captures their motion with 12 evenly-surround cameras on diverse 3D UE platforms. Extensive experiments show that 3DTrajMaster sets a new state-of-the-art in both accuracy and generalization for controlling multi-entity 3D motions. Project page: http://fuxiao0719.github.io/projects/3dtrajmaster.

## 1 Introduction

Controllable video generation (Brooks et al., 2024; Guo et al., 2023b; Chen et al., 2023) aims to synthesize high-fidelity videos that are controlled by user inputs, such as text prompts, sketches, or bounding boxes. A critical objective in controllable video generation is the precise manipulation of object motions within videos, which is essential for simulating the dynamic world and potentially aids video generative models in understanding the underlying physics of the world. In addition, it can unleash many applications of video generative models, such as virtual cinematography for the film industry, acting as interactive games, and providing world models for embodied AI systems.

Recently, there has been some methods attempting to manipulate object motions in video generation by introducing 2D control signals, such as 2D sketches (Wang et al., 2024b; Guo et al., 2023a), bounding boxes (Yang et al., 2024; Wang et al., 2024a), and points (Wang et al., 2024c; Zhang et al., 2024). These methods offer convenient user interactions and have delivered impressive video generation results. However, we argue that 2D control signals cannot fully express the inherent 3D nature of motion, which limits the control capability of object motions. As real-world objects move in 3D space, some motion properties can only be described through 3D representations. For example, the rotation of an object can be succinctly described using three parameters in 3D, and occlusions between objects can be simply represented using z-buffering. In contrast, it is quite difficult for 2D control signals to represent these concepts.

In this paper, we focus on the problem of controlling multi-object 3D motions in video generative models, aiming to simulate the authentic dynamics of objects in 3D space. This setting is

---

†: Work done during an internship at KwaiVGI, Kuaishou Technology. ✉: Corresponding Authors.

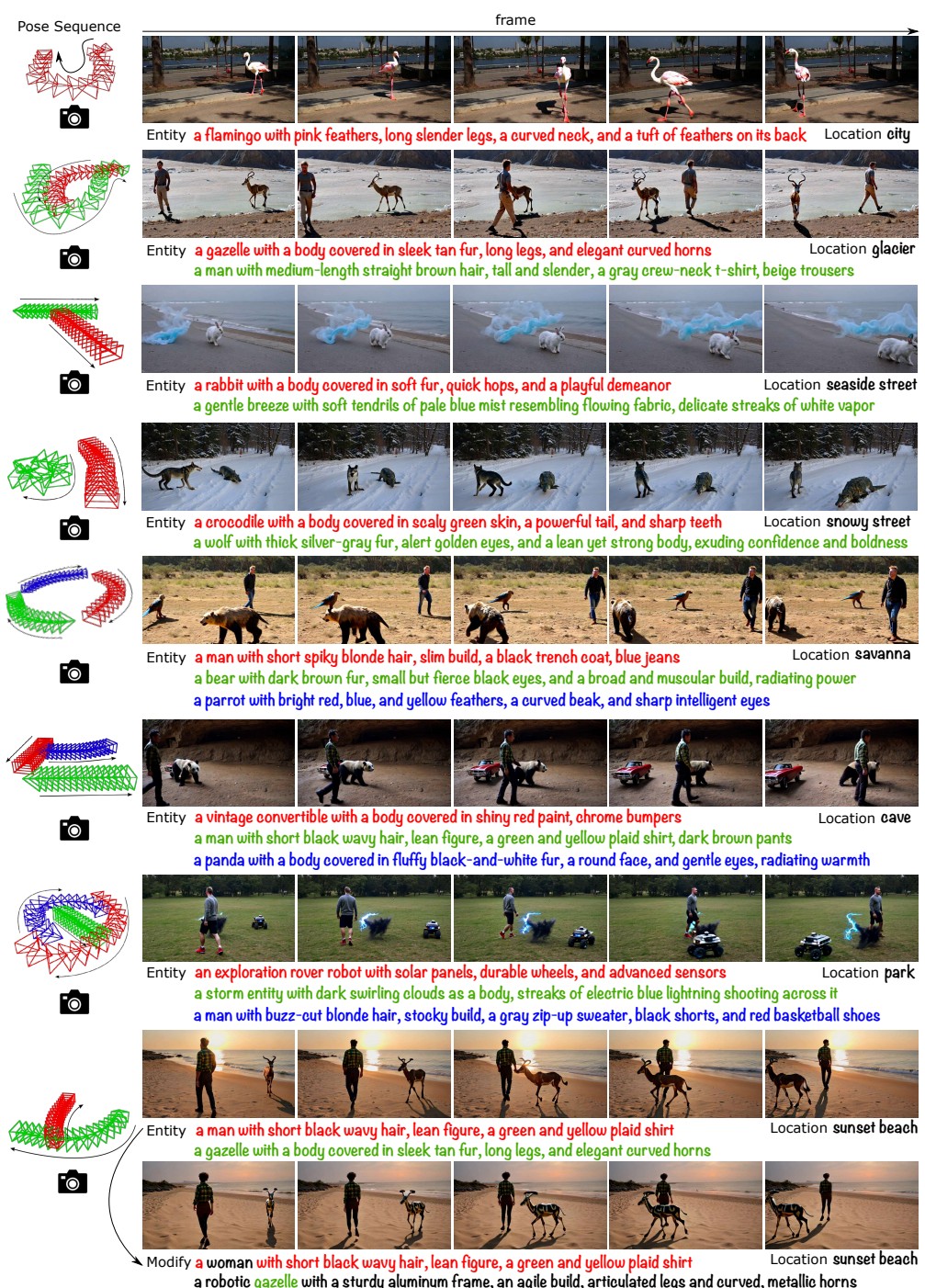

Figure 1: **3DTrajMaster controls one or multiple entity motions in 3D space with input entity-specific 3D trajectories for text-to-video (T2V) generation.** It allows diverse entity categories (human, animal, car, robot, natural force, etc) and flexible edits on entity descriptions (see more in Fig. S11). The text prompt is "*{Entity 1},..., and {Entity N} is/are moving in the {Location}*". *(We kindly urge readers to check more generalizable results (≥200) in our website)*

more aligned with the requirements of downstream applications, such as emulating realistic human motions in movies or exploring 3D virtual scenes in games. However, this problem is extremely challenging. There are three core questions we need to answer: 1) How to precisely represent the 3D motions of objects; 2) How to correlate multiple object descriptions with their respective motion

sequences in video generative models; 3) How to maintain the generalization capability of video models after injecting 3D motion information.

To address these, we propose a novel approach, **3DTrajMaster**, which is able to manipulate multi-entity motions in 3D space for video generation by leveraging entity-specific 6DoF pose sequences as additional inputs. The core of our model is a plug-and-play 3D-motion grounded object injector, which associates each entity with their corresponding pose sequences, and then injects these conditions into the foundation model, to control the entity motion. Specifically, the entities and trajectories are projected into latent embeddings via a frozen text encoder and a learnable pose encoder, respectively. These two modality embeddings are then entity-wise added to form correspondences, which are further fed into a gated self-attention layer for motion fusion. This plug-and-play architecture preserves the video model's prior and can generalize on more diverse entities and 3D trajectories.

However, another challenge in training our model lies in data availability. Existing video datasets face two key limitations: 1) *Low entity diversity:* Datasets with paired entities and 3D trajectories are mostly limited to humans and autonomous vehicles, with inconsistent spatial distributions and overcrowded entities. 2) *Inaccurate/Failed pose estimation:* Current 6D pose estimation methods focus on rigid objects, while non-rigid objects, such as animals, are underrepresented, with only human poses studied using SMPL (Loper et al., 2023). To this end, we choose to construct a custom dataset, termed 360°-Motion Dataset, with unified trajectory distribution using advanced UE rendering techniques. We start by collecting 3D assets of humans and animals and rescaling them to a unified cubic space. GPT (Achiam et al., 2023) is then employed to generate 3D trajectory templates for these assets. Various entities and trajectory templates are arranged and combined to create diverse motions. These globally animated assets are captured using 12 evenly positioned cameras within the collected 3D scenes, including city (MatrixCity (Li et al., 2023a)), desert, forest, and HDRIs (projected into 3D space)[1]. To prevent video domain shift in our constructed dataset, we introduce two key components: 1) A video domain adaptor, which is trained to fit data distribution and slightly reduced during inference. 2) An annealed sampling strategy, where trajectories are injected to guide general motion in the early steps and drop out in the later stages.

We evaluate our 3DTrajMaster in the curated novel pose sequences with GPT-generated entity prompts, obtaining a significant lead over current SOTAs. In summary, our contributions are:

1) We are the first to customize 6 degrees of freedom (DoF) multi-entity motion in 3D space for controllable video generation, establishing a new benchmark for fine-grained motion control.

2) We propose a 3D-motion grounded video diffusion model that controls multi-entity motions using pose sequences as motion representations. Our flexible object injector enforces entity-wise correspondence between objects and their motions and preserves the video diffusion prior.

3) We introduce a scalable 4D motion dataset construction mechanism, and techniques like the video domain adaptor and annealed sampling to enhance video quality while maintaining motion accuracy.

4) 3DTrajMaster achieves state-of-the-art accuracy in controlling 3D entity motions and allows fine-grained entity input customization such as changing human hair, clothing, gender, and figure size.

## 2 RELATED WORK

**Customizing Video Motion with 2D Guidance.** Previous methods predominantly perform motion control on 2D spaces, as this aligns more easily with the input video format. A straightforward path is to direct videos based on motion patterns from reference videos (Zhao et al., 2023; Jeong et al., 2024; Ling et al., 2024). However, they require users to provide reference video templates. While training-free paradigms (Yang et al., 2024; Xiao et al., 2024), utilizing attention mechanisms to edit spatial-temporal layouts, can mitigate this issue, they exhibit poor generalization in real-world scenarios and rely heavily on trial-and-error. Further advancements utilize more high-level representations, such as sketches&depths (dense or sparse) (Wang et al., 2024b; Guo et al., 2023a), pose skeletons (Feng et al., 2023; Xu et al., 2024; Chen et al., 2024), bounding boxes (Wang et al., 2024a), and 2D trajectories (Wang et al., 2024c; Zhang et al., 2024; Yin et al., 2023; Yang et al., 2024), to enable more flexible motion generation. Although these methods can model camera, object, or joint movements, the lack of 3D awareness limits precise 3D motion control.

---

[1]Poly Haven: https://polyhaven.com/

**Learning 3D-aware Motion Synthesis.** Considering that video is a sequence of images projected from 3D world, manipulating video in 3D space is both more crucial and impactful. A key aspect of this manipulation is camera movement. MotionCtrl (Wang et al., 2024c) is the first to regulate video using camera poses (rotation and translation) in 3D space, while CameraCtrl (He et al., 2024) and VD3D (Bahmani et al., 2024b) further enhance camera representation with plücker embeddings (Sitzmann et al., 2021). SynCamMaster (Bai et al., 2024) extends single-camera control to multi-camera synchronization. GameGen-X (Che et al., 2024) can generate game videos with novel 'WASD' keyboard inputs. Other approaches (Hou et al., 2024; Hu et al., 2024a) also explore training-free paradigms. However, none address the customization of object motion in 3D space. Manipulation on 2D maps (Wang et al., 2024c; Zhang et al., 2024) often fails in multi-object scenarios, particularly with 1) aligning each entity and its corresponding motion, 2) handling *3D occlusion*. In contrast, 3DTrajMaster is the first to overcome them and simulate plausible 3D motions.

## 3    3DTRAJMASTER

Our goal is to master entity motions in 3D space for text-to-video (T2V) generation by leveraging entity-specific 3D trajectories as additional inputs. To this end, we introduce *3DTrajMaster* (see Fig. 2), a 3D-motion grounded video diffusion model trained in two stages. First, we describe the video diffusion model and the task formulation (Sec. 3.1). Then, we present our proposed model, whose core is to train a plug-and-play 3D grounded object injector to integrate multiple detailed entity descriptions and the respective pose sequences (Sec. 3.2). We further incorporate a domain adaptor to mitigate video domain shifts introduced by our constructed training data (Sec. 3.3). Finally, we detail the inference process using annealed sampling to enhance video quality (Sec. 3.4).

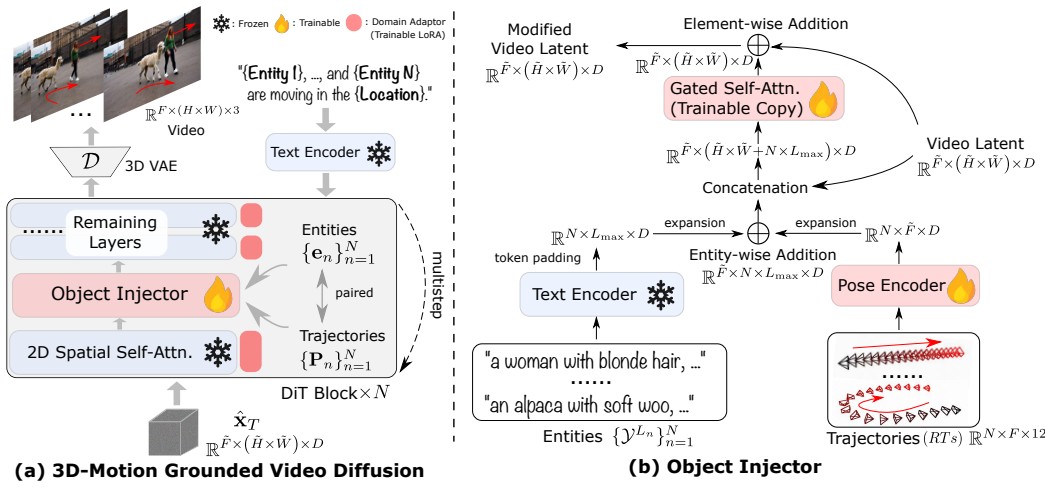

Figure 2: **3DTrajMaster Framework.** Given a text prompt consisting of N entities $\{\mathbf{e}_n\}_{n=1}^N$, 3DTrajMaster (a) is able to generate the desired video with entity motions that conform to the input entity-wise pose sequences $\{\mathbf{P}_n\}_{n=1}^N$. Specifically, it involves two training phases. First, it utilizes a domain adaptor to mitigate the negative impact of training videos. Then, an object injector module is inserted after the 2D spatial self-attention layer to integrate paired entity prompts and 3D trajectories. (b) Details of the object injection process. The entities are projected into latent embeddings through the text encoder. The paired pose sequences are projected using a learnable pose encoder and then fused with entity embeddings to form entity-trajectory correspondences. This condition embedding is concatenated with the video latent and fed into a gated self-attention layer for motion fusion. Finally, the modified latent gets back to the remaining layers in the DiT block.

### 3.1    PRELIMINARIES ON 3D-ENTITY-AWARE VIDEO DISTRIBUTION

**Video Diffusion Models.** Latent text-to-video diffusion model (Ho et al., 2022a;b; Brooks et al., 2024; Chen et al., 2023; Blattmann et al., 2023) learns the conditional distribution $p(\mathbf{x}|\mathbf{c})$ of encoded video data $\mathbf{x}$ ($\mathbf{x} = \mathcal{E}(X)$, $\mathcal{E}(\cdot)$ is VAE encoder) given text description $\mathbf{c}$ in latent space. In the forward progress, it progressively transits the clean data $\mathbf{x}_0$ to the desired Gaussian distribution in a Markov

chain: $\{\mathbf{x}_t, t \in (1, T) \mid \mathbf{x}_t = \alpha_t \mathbf{x}_0 + \sigma_t \boldsymbol{\epsilon}, \boldsymbol{\epsilon} \sim \mathcal{N}(\mathbf{0}, \mathbf{I})\}$. To iteratively recover the data $\hat{\mathbf{x}}_0$ from the noise $\boldsymbol{\epsilon} \sim \mathcal{N}\left(\mathbf{0}, \sigma_t^2 \mathbf{I}\right)$, it learns a denoising model $\hat{\boldsymbol{\epsilon}}_{\boldsymbol{\theta}}$ with the objective function: $\boldsymbol{\epsilon} \approx \hat{\boldsymbol{\epsilon}}_{\boldsymbol{\theta}}(\mathbf{x}_t; t, \mathbf{c})$. With the preconditioning strategy (Karras et al., 2022; Salimans & Ho, 2022), it optimizes the neural network $\hat{F}_{\boldsymbol{\theta}}$ by parameterizing the $\hat{\boldsymbol{\epsilon}}_{\boldsymbol{\theta}}$ as: $\hat{\boldsymbol{\epsilon}}_{\boldsymbol{\theta}} = c_{\text{out}}(\sigma_t) \hat{F}_{\boldsymbol{\theta}}\left(c_{\text{in}}(\sigma_t) \mathbf{x}_t; \mathbf{c}, \sigma_t\right) + c_{\text{skip}}(\sigma_t) \mathbf{x}_t$.

**Task Formulation.** Given an input text prompt $\mathbf{c}$ consisting of N entities $\{\mathbf{e}_n\}_{n=1}^N$ and their paired 3D trajectories $\{\mathbf{P}_n\}_{n=1}^N$, where $\mathbf{P}_n^f = [\mathbf{R}; \mathbf{T}] \in \mathbb{R}^{3 \times 4}$ for $f$-th frame and object orientation and translation are represented by $\mathbf{R} \in \mathbb{R}^{3 \times 3}$ and $\mathbf{T} \in \mathbb{R}^3$, respectively, our goal is to generate plausible video $\mathbf{X} \in \mathbb{R}^{F \times H \times W}$ that accords with each entity description $\mathbf{e}$ and the respective trajectory $\mathbf{P}$. The overall generative formulation $f(\cdot)$ is

$$f(\cdot) : \mathbf{c} \in \mathcal{Y}^L, (\mathbf{e}_n \in \mathcal{Y}^{L_n}, \mathbf{P}_n \in \mathbb{R}^{3 \times 4})_{n=1}^N \to \mathbf{X} \in \mathbb{R}^{F \times H \times W} \tag{1}$$

where $\mathbf{X} \approx \mathcal{D}(\hat{\mathbf{x}}_0)$ ($\mathcal{D}(\cdot)$ is the VAE decoder), $\hat{\mathbf{x}} = p(\hat{\mathbf{x}}_T) \prod_{t=1}^T p_{\boldsymbol{\theta}}\left(\hat{\mathbf{x}}_{t-1} \mid \hat{\mathbf{x}}_t, \mathbf{c}, (\mathbf{e}_n, \mathbf{P}_n)_{n=1}^N\right)$, $\mathcal{Y}$ is the alphabet, and $L$ is the token length. Our primary challenge lies in modeling the distribution $p_{\boldsymbol{\theta}}$ or specifically $\hat{\boldsymbol{\epsilon}}_{\boldsymbol{\theta}}$ to generate realistic videos that accurately correspond to the given multiple 3D entity conditions. Here we structure $\hat{\boldsymbol{\epsilon}}_{\boldsymbol{\theta}}(\mathbf{x}; \mathbf{c}, \sigma_t, (\mathbf{e}_n, \mathbf{P}_n)_{n=1}^N)$ as transformer architecture (Peebles & Xie, 2023) for its superior scalability and performance over U-Net (Ronneberger et al., 2015).

## 3.2 PLUG-AND-PLAY 3D-MOTION GROUNDED OBJECT INJECTOR

**Matching Entity-Trajectory Pair.** The entity prompts $\{\mathbf{e}_n\}_{n=1}^N$ are projected into latent embeddings $\{\mathbf{Z}_n^{\mathbf{e}}\}_{n=1}^N$ using a frozen text encoder $\mathcal{E}_{\mathbf{T}}(\cdot) : \mathbf{e}_n \in \mathcal{Y}^{L_n} \to \mathbf{Z}_n^{\mathbf{e}} \in \mathbb{R}^{L_{\max} \times D}$, where each embedding $\mathbf{Z}_n^{\mathbf{e}}$ is zero-padded to maximum token length $L_{\max}$. Correspondingly, the pose sequences $\{\mathbf{P}_n\}_{n=1}^N$ are also projected into latent embeddings $\{\mathbf{Z}_n^{\mathbf{P}}\}_{n=1}^N$ through the trainable pose encoder $\mathcal{E}_{\mathbf{P}}(\cdot) : \mathbf{P}_n \in \mathbb{R}^{F \times 12} \to \mathbf{Z}_n^{\mathbf{P}} \in \mathbb{R}^{\tilde{F} \times D}$. The pose encoder $\mathcal{E}_{\mathbf{P}}$ consists of a linear layer and a downsampler along the temporal dimension, resembling the causal encoding applied to video input $\mathbf{x}$ in 3D VAE, where the mapping function is $\mathcal{E}_{\mathbf{X}}(\cdot) : \mathbf{X} \in \mathbb{R}^{F \times H \times W} \to \mathbf{x} \in \mathbb{R}^{\tilde{F} \times \tilde{H} \times \tilde{W}}$. Here the downsampler refers to interval sampling of tensors, where we also tried several sequential one-dimensional convolution layers but achieved similar results. Then, the paired entity and trajectory embeddings are expanded and combined through entity-wise addition to form a bonded entity-motion correspondence $\mathbf{Z}^{\mathbf{Pe}} \in \mathbb{R}^{\tilde{F} \times N \times L_{\max} \times D}$.

**Gated Self-Attention for Motion Fusion.** Inspired by (Li et al., 2023b), we employ a gated self-attention layer to handle multiple entity-trajectory pairs $\mathbf{Z}^{\mathbf{Pe}}$ (with varying dimensional embeddings) as input, while further refining the correlated features. Specifically, we replicate the weight of the 2D spatial self-attention layer in each DiT block as initialization to enable grounding. The input video tokens $\mathbf{x}_t$ and $\mathbf{Z}^{\mathbf{Pe}}$ are passed through this trainable copy via truncated self-attention. The output can be expressed in a residue-connection form:

$$\begin{aligned} \mathbf{x}_t &= \mathbf{x}_t + \beta \cdot \mathbf{Tc}(\mathbf{Att}(\mathbf{q}, \mathbf{k}, \mathbf{v})) \\ \mathbf{q} &= \mathbf{Q} \cdot \mathbf{T}, \mathbf{k} = \mathbf{K} \cdot \mathbf{T}, \mathbf{v} = \mathbf{V} \cdot \mathbf{T}, \mathbf{T} = \mathbf{x}_t \oplus \mathbf{Z}^{\mathbf{Pe}} \end{aligned} \tag{2}$$

where $\beta$ is a trainable scale, $\mathbf{Tc}(\cdot)$ is the truncation operation to preserve $\mathbf{x}_n$ tokens, $\mathbf{Att}(\cdot)$ is softmax attention, $\mathbf{Q}$, $\mathbf{K}$ and $\mathbf{V}$ are query, key and value embedding matrices, and $\oplus$ denotes concatenation. In this stage, we train the $\boldsymbol{\theta}_1$ including the pose encoder and the gated self-attention parameters as follow.

$$\mathcal{L}(\boldsymbol{\theta}_1) = \mathbb{E}_{\mathbf{x}, \mathbf{c}, \boldsymbol{\epsilon} \sim \mathcal{N}(\mathbf{0}, \sigma_t^2 \mathbf{I}), \mathbf{e}, \mathbf{P}, t, \beta}\left[\left\|\boldsymbol{\epsilon} - \hat{\boldsymbol{\epsilon}}_{\boldsymbol{\theta}_1}\left(\mathbf{x}_t, \mathbf{c}, (\mathbf{e}_n, \mathbf{P}_n)_{n=1}^N\right), t, \beta\right\|_2^2\right] \tag{3}$$

## 3.3 ALLEVIATING VIDEO DOMAIN SHIFT FROM CONSTRUCTED TRAINING DATA

**360°-Motion Dataset.** High-quality training data is vital for learning generalizable 3D motion control. A straightforward preparation is to extract paired entity descriptions and 6DoF poses from common video datasets. However, it is hard due to twofold: 1) *Low diversity/quality entity*: Datasets with paired entities and 3D trajectories are mostly limited to humans (Jiang et al., 2024; Araújo et al., 2023) and autonomous vehicles (Geiger et al., 2012; Sun et al., 2020), where the spatial distributions vary between datasets and the entities may be overcrowded. In video datasets like Artgrid, Pixabay, and Pexels[2], human category occupies a relatively large proportion in 3D/4D asset

---

[2]Artgrid: https://artgrid.io/, Pixabay: https://www.videvo.net/, Pexels: https://www.pexels.com/

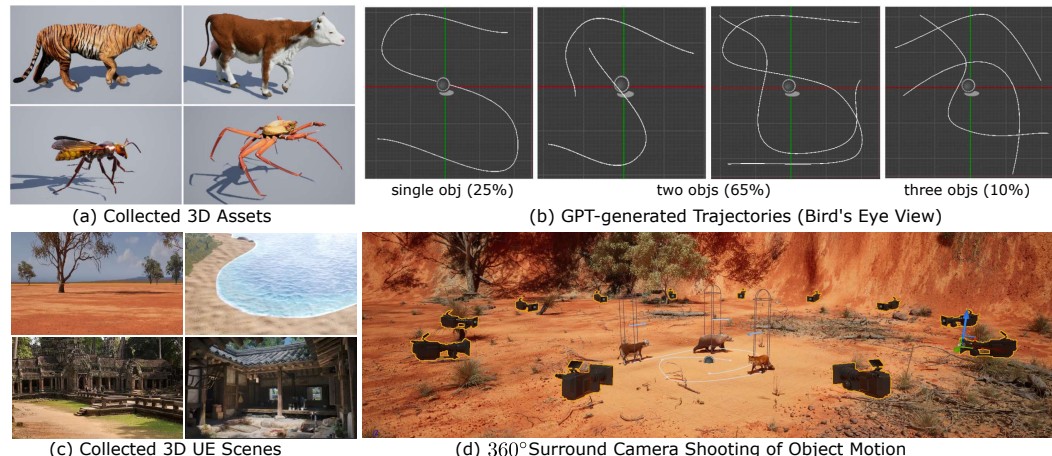

Figure 3: **Dataset Construction Illustration.** We correlate (a) collected 3D assets with (b) GPT-generated 3D trajectories on (c) diverse 3D UE platforms, positioning (d) 12 evenly distributed surrounding cameras to capture the object motions in video format.

objectives (refer to Sec. E.2), limiting model generalization to other categories like animals and vehicles. Issues like watermarks in WebVid (Bain et al., 2021) further increase the cost of filtering. 2) *Low-accuracy/Failed pose estimation*: Most 6D pose estimation methods exclusively focus on rigid objects, and rely on CAD models (Labbé et al., 2022; Wen et al., 2024) or posed multi-view images (Liu et al., 2022; Sun et al., 2022). For non-rigid animated objects, only human poses have been widely studied via methods like SMPL (Loper et al., 2023), limiting the estimation for general 4D objects, such as animals. A simpler alternative is to represent only 3D locations via depth models (Hu et al., 2024b; Ke et al., 2024; Fu et al., 2025). However, there exist errors in segmenting the foreground entities from the background and can not generate consistent video metric depth.

To circumvent the aforementioned challenges, we opt to construct a synthetic dataset, named *360°-Motion*, through Unreal Engine (UE) with advanced rendering technologies (see Fig. 3). We begin by collecting 70 animated 3D assets across two categories: human and animal. Humans are differentiated by attributes such as gender, clothing, body shape, and hairstyle. GPT-4V (OpenAI, 2023) is then used to generate text descriptions $\mathbf{e}_n \in \mathcal{Y}^{L_n}(L_n \leq 20)$ for each rendered asset image (Fig. 3 (a)). For posed object trajectory templates (Fig. 3 (b)), we follow TC4D (Bahmani et al., 2024a) by leveraging GPT to generate 3D spline (location $\mathbf{T}$) and additional orientation $\mathbf{R}$ via the gradient calculation on spline. This process yields approximately 96 templates in canonical space, each associated with one to three assets. We additionally reduce the size of the animals by a ratio of 0.6 to prevent collisions with other assets. The paired assets and their motion templates are then placed within a 5×5 square meter range in one of the 3D platforms, including city (MatrixCity (Li et al., 2023a)), dessert, forest, and HDRIs (projected into 3D). We position 12 sets of cameras evenly around the scene to capture 360-degree views, producing 100 frames per video clip at 384×672 resolution for each camera. This process produces a total of 54,000 videos by arranging and combining various objects and trajectories. (see Sec. E.1 and Supp. video samples for illustration)

**Video Domain Adaptor.** Training video diffusion models on this relatively small set of constructed video clips can lead to an undesirable UE style, limiting the generalization ability. To prevent learning this variation in quality and retain the knowledge of the base T2V, we train LoRA modules (Hu et al., 2021) that serve as video domain adaptor. Specifically, we integrate LoRA into self-attention, cross-attention, and linear layers of the base T2V model, as shown in Fig. 2. The attention/linear projection matrices $\{\mathbf{W}_n\}_{n=1}^{K}$ are associated with additional trainable lower rank matrices $\{\Delta\mathbf{W}_n = \alpha\mathbf{A}_n\mathbf{B}_n^T\}_{n=1}^{K}$, where $\alpha$ is the scaler that can be adjusted to control the adaptor influence. During inference, we set $\alpha$ to a small value to mitigate the negative impact of synthetic video data. We optimize $\boldsymbol{\theta_2} = \{\Delta\mathbf{W}_n\}_{n=1}^{K}$ with the training objective:

$$\mathcal{L}(\boldsymbol{\theta_2}) = \mathbb{E}_{\mathbf{x},\mathbf{c},\boldsymbol{\epsilon}\sim\mathcal{N}(\mathbf{0},\sigma_t^2\mathbf{I}),t,}\left[\|\boldsymbol{\epsilon} - \hat{\boldsymbol{\epsilon}}_{\boldsymbol{\theta_1}}(\mathbf{x}_t,\mathbf{c},t,\alpha)\|_2^2\right]. \tag{4}$$

Note that the domain adaptor $\boldsymbol{\theta_2}$ is frozen when training the object injector $\boldsymbol{\theta_1}$.

## 3.4 Inference Procedure

We initialize the video latent $\hat{\mathbf{x}}_T$ as standard Gaussian noise, and progressively denoise it with the guidance of desired entity-trajectory pairs $(\mathbf{e}_n, \mathbf{P}_n)_{n=1}^N$, following the same schedule as the previous two training stages. We apply classifier-free guidance (Ho & Salimans, 2022) and use DDIM (Song et al., 2020) for re-spaced sampling for acceleration. To further enhance the video quality, we employ an annealed sampling strategy (Algorithm 1): During inference in the former steps, trajectories are inserted into the model to define the general object motions, while in the latter stage, they are dropped out, transitioning to the standard T2V generation process. We also observe that setting negative 3D trajectories as static motions $\{(\hat{\mathbf{P}}_n)_{n=1}^N | \hat{\mathbf{P}}_n = \mathbf{P}_0, \forall n\}$ can further improve pose accuracy. This phenomenon reflects the model's ability to learn 3D motion representations: Since we do not randomly drop out motion sequences during training like text, the model implicitly learns static motion modeling from videos where entities are primarily in motion. Thus when setting static motion as a "negative motion prompt", we can amplify the magnitude of entity movement, leading to improved pose accuracy during evaluation. However, we do not adopt it as it sometimes results in a video quality decline (refer to Sec. F.2.2).

---

**Algorithm 1** Annealed conditional sampling with classifier-free guidance (CFG)

---

**Require:** $w$: guidance strength, $T_c$: annealed timestep, $\alpha$: LoRA modulator, $\tilde{\boldsymbol{\theta}}$: frozen base T2V model, $\boldsymbol{\theta}_1$: object injector, $\boldsymbol{\theta}_2$: domain adaptor, $\mathbf{c}$: text condition, $(\mathbf{e}, \mathbf{P})$: entity-trajectory pairs

1: $\hat{\mathbf{x}}_1 \sim \mathcal{N}\left(\mathbf{0}, \sigma_t^2 \mathbf{I}\right)$
2: **for** $t = 1, ..., T$ **do**
3:     **if** $\leq T_c$ **then**
4:         $\tilde{\boldsymbol{\epsilon}}_t = (1 + w)\hat{\boldsymbol{\epsilon}}_{\tilde{\boldsymbol{\theta}}, \boldsymbol{\theta}_1, \boldsymbol{\theta}_2}\left(\hat{\mathbf{x}}_t, \mathbf{c}, (\mathbf{e}_n, \mathbf{P}_n)_{n=1}^N, \alpha\right) - w\hat{\boldsymbol{\epsilon}}_{\tilde{\boldsymbol{\theta}}, \boldsymbol{\theta}_1, \boldsymbol{\theta}_2}\left(\hat{\mathbf{x}}_t, \alpha\right)$
5:     **else**
6:         $\hat{\boldsymbol{\epsilon}}_t = (1 + w)\boldsymbol{\epsilon}_{\tilde{\boldsymbol{\theta}}}\left(\hat{\mathbf{x}}_t, \mathbf{c}\right) - w\boldsymbol{\epsilon}_{\tilde{\boldsymbol{\theta}}}\left(\hat{\mathbf{x}}_t\right)$
7:     **end if**
8:     $\hat{\boldsymbol{z}}_t = \left(\hat{\mathbf{x}}_t - \sigma_t \tilde{\boldsymbol{\epsilon}}_t\right)/\alpha_t$
9:     $\hat{\mathbf{x}}_{t+1} \sim \mathcal{N}\left(\hat{\mathbf{x}}_{t+1}; \tilde{\boldsymbol{\mu}}_{t+1|t}\left(\hat{\boldsymbol{z}}_t, \hat{\mathbf{x}}_t\right), \sigma_{t+1|t}^2 \mathbf{I}\right)$ if $t < T$ else $\hat{\mathbf{x}}_{t+1} = \hat{\boldsymbol{z}}_t$
10: **end for**
11: **return** $\hat{\mathbf{x}}_{t+1}$

---

## 4 Experiments

### 4.1 Implementation Details

For input text prompts, we use a unified template: "*{Entity 1},..., and {Entity N} are moving in the {Location}*." Here we set "{Location}" based on the respective 3D UE platform. We train 3DTrajMaster based on our internal video diffusion model for research purposes (see Sec. A for more details), which contains $\sim$ 1B parameters. The clipped training video and inference video are set to $384 \times 672$ resolutions. Each video segment is 5 seconds long. We utilize the Adam optimizer and train on a cluster of 8 NVIDIA H800 GPUs, with a learning rate of $5 \times 10^{-5}$ and a batch size of 8. The training process consisted of 50,000 steps for the domain adaptor and an additional 36,000 steps for the object injector. During inference, we set the DDIM steps as 50 and the CFG as 12.5.

### 4.2 Baselines

We compare 3DTrajMaster with existing SOTA methods that are capable of customizing object motions: MotionCtrl (Wang et al., 2024c), Direct-a-Video (Yang et al., 2024) and Tora (Zhang et al., 2024). We configure these baseline models using their best performance settings, based on their official open-sourced codebases.

### 4.3 Evaluation Metric

1) *Trajectory accuracy*: Due to the absence of a pose estimator for open-world 4D objects, we limit our evaluation to only human objectives. Specifically, we utilize GVHMR (Shen et al., 2024) to estimate human poses $\{(\mathbf{R}_n^{est}, \mathbf{T}_n^{est})\}_{n=1}^F$ and compare them with the input pose sequences

$\{(\mathbf{R}_n^{gt}, \mathbf{T}_n^{gt})\}_{n=1}^F$. We align the two trajectories at the first frame location. We follow CameraCtrl (He et al., 2024) to estimate the rotation angle error **RotErr** and translation scale error **TransErr**, but take the average rather than the sum. 2) *Video quality*: We leverage standard metrics such as Frechét Video Distance (**FVD**) (Unterthiner et al., 2018), Frechét Image Distance (**FID**) (Seitzer, 2020), and CLIP Similarity (**CLIPSIM**) (Wu et al., 2021) to assess the video appearance.

## 4.4 EVALUATION DATASET

1) *Pose Sequence*: We collect 44 novel pose templates, each comprising one or more object motions. 2) *Entity Description*: we use GPT to generate 20 novel human, 52 novel non-human descriptions, and 32 novel locations (refer to Sec. E.3), which are randomly assigned to poses to form 100 pairs (12 single-entity, 72 two-entity, and 16 three-entity each pair has one human entity).

## 4.5 COMPARISON

**Granularity Level.** As shown in Table 1, 3DTrajMaster can customize object location and orientation in 3D space. In contrast, 2D motion representations such as points (MotionCtrl/Tora) and bounding box (Direct-a-Video), lack awareness of the z dimension. This ambiguity becomes more problematic when handling 3D occlusion. Besides, MotionCtrl and Tora integrate multiple entities into a single 2D feature, lacking the capability to correlate individual entities with their respective trajectories (see failure case in Fig. 6). When tested on multi-entity input, Direct-a-Video (a training-free paradigm) shows particularly weak results. Furthermore, 3DTrajMaster allows for diverse entities and backgrounds (see Fig. 4), and detailed control of entity inputs (see Fig. 5).

Table 1: **Fine Control Comparison with Multi-Entity Input.**

|  | Location | Orientation | Entity-Traj. Corresp. | Learning-based? |
|---|---|---|---|---|
| Direct-a-Video | ✓(2D) | ✗ | ✓ | ✗ |
| MotionCtrl/Tora | ✓(2D) | ✗ | ✗ | ✓(not decoupled) |
| 3DTrajMaster (Ours) | ✓ (3D) | ✓ | ✓ | ✓(decoupled) |

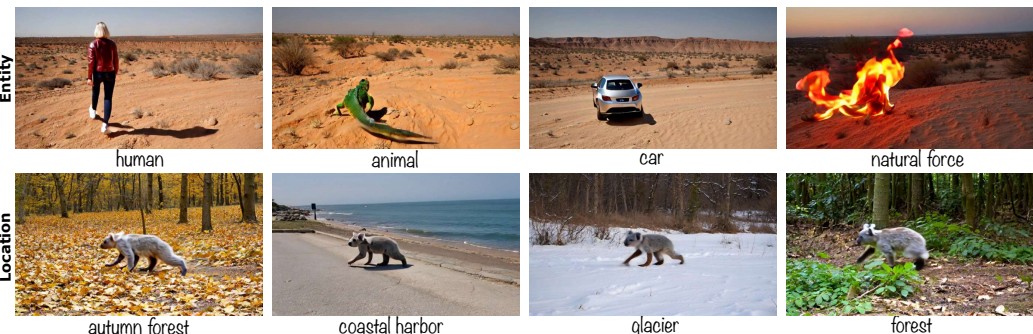

Figure 4: **Diversity on Entity and Background.** 3DTrajMaster can control versatile entities (human, animal, car, robot, and even abstract natural force), while also generating diverse locations.

Input human entity "a man with short black hair, medium figure, gray striped sweater, black jeans, light brown leather shoes"

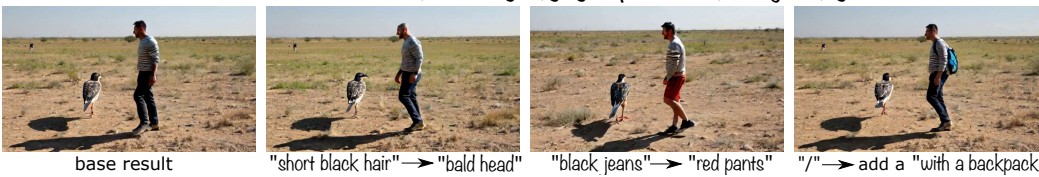

base result     "short black hair"→"bald head"     "black jeans"→"red pants"     "/"→add a "with a backpack"

Figure 5: **Fine-grained Editing on Human Entity Input.** 3DTrajMaster supports modifications in attributes such as hair, clothing, figure size, and so on. (Please check more in Fig. S11)

**Quantative & Qualitative Results.** To align with the input requirement of MotionCtrl and Direct-a-Video, we project the 3D pose trajectories onto 2D space. For baselines, we simplify the entity

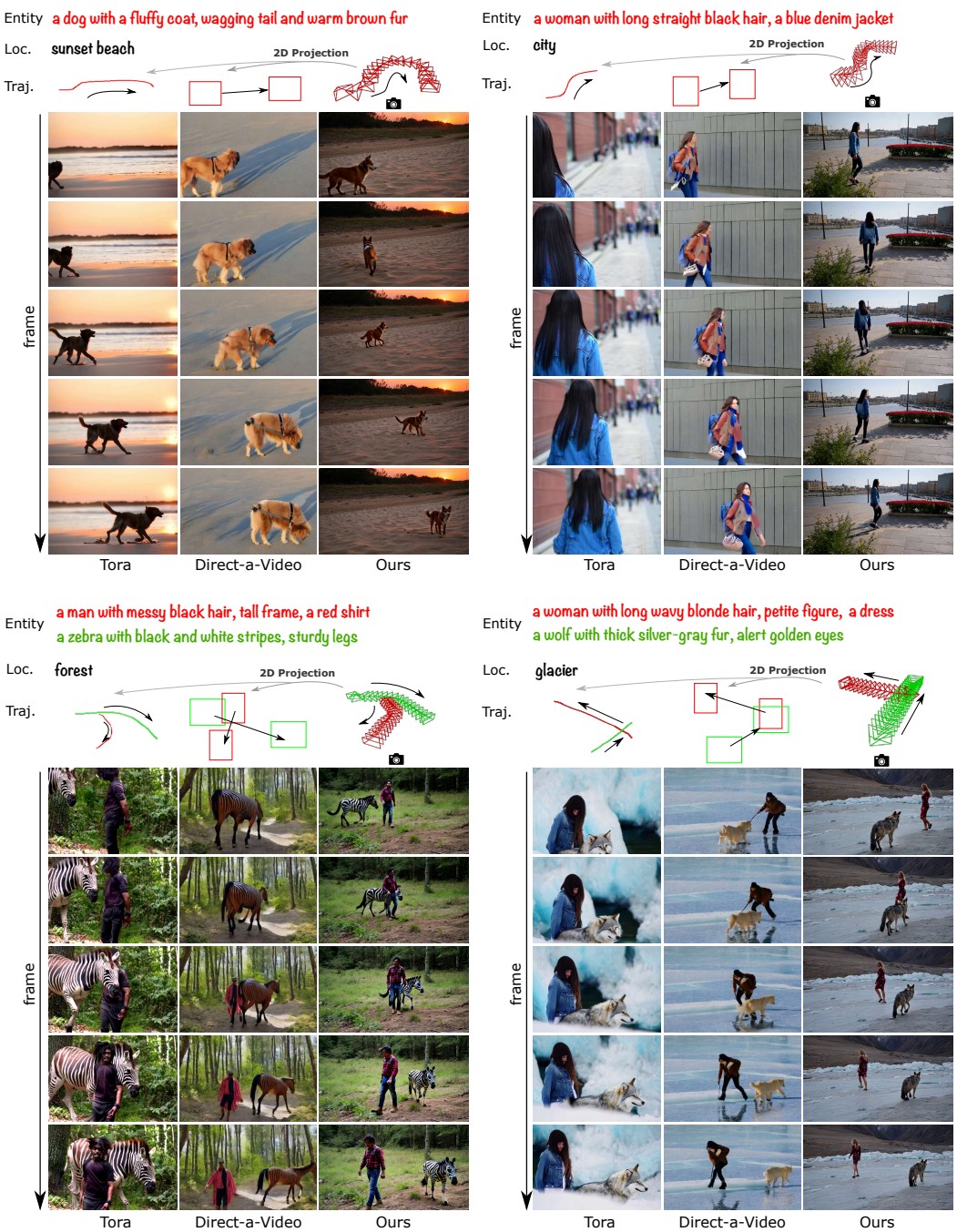

Figure 6: **Qualitative Comparison on Single/Multiple Entity Motion.** 3DTrajMaster outperforms all 2D baselines by modeling 6 DoF entity motion, which can better express the inherent 3D nature of motion. In the last figure, Tora mistakenly regards the background entity as the girl entity.

description, such as changing "a man with messy black hair, tall frame, a red shirt" to "a man" or "a man in red". Otherwise, they may fail to generate videos with detailed descriptions. As shown in Fig. 6, in single entity settings, 3DTrajMaster generates precise entity motion, such as a 180° turn-back and a continuous inward 90° turn-around. In contrast, Tora and Direct-a-Video produce simpler motions, merely shifting objects from left to right or top-right. In the multi-entity benchmark, 3DTrajMaster successfully handles 3D occlusions, such as a man walking in front of a

Table 2: **Quantative Comparison on Single/Multiple Entity Motion.** 3DTrajMaster performs better on multiple entity input since the single entity trajectory is more complex.

| Methods | Single Entity | | Multiple Entities | | All Entities | |
|---|---|---|---|---|---|---|
| | TransErr (m) ↓ | RotErr (deg) ↓ | TransErr (m) ↓ | RotErr (deg) ↓ | TransErr (m) ↓ | RotErr (deg) ↓ |
| Base T2V | 1.946 | 1.799 | 1.586 | 1.208 | 1.629 | 1.279 |
| MotionCtrl | 1.752 | 2.134 | 1.682 | 1.613 | 1.690 | 1.675 |
| Tora | 1.707 | 1.158 | 1.867 | 1.514 | 1.848 | 1.471 |
| Direct-a-Video | 1.632 | 1.902 | 1.391 | 0.942 | 1.420 | 1.057 |
| 3DTrajMaster | **0.456** | **0.319** | **0.390** | **0.272** | **0.398** | **0.277** |

zebra. Direct-a-Video, however, fails in overlapping regions with mixed man and zebra. We report metric results in Table 2. It is not surprising that ours significantly outperforms all baselines.

## 4.6 ABLATION STUDY

Table 3: **Ablation Study on Full Testest and Base T2V Videos (As Reference Video).**

| Ablation Setting | Video Quality | | | 3D Trajectory Accuracy | |
|---|---|---|---|---|---|
| | FVD ↓ | FID ↓ | CLIPSIM ↑ | TransErr (m) ↓ | RotErr (deg) ↓ |
| w/ Cross-Attn. Fusion | 1673.24 | 102.13 | 32.87 | 0.453 | 0.341 |
| w/ 3D Self-Attn. | 1597.51 | 98.74 | 33.15 | 0.427 | 0.296 |
| w/o Domain Adaptor | 2379.89 | 157.51 | 30.50 | 0.415 | 0.301 |
| w/o Annealed Sampl. | 1841.64 | 112.57 | 32.26 | 0.407 | **0.265** |
| Full Model | **1546.15** | **96.75** | **33.77** | **0.398** | 0.277 |

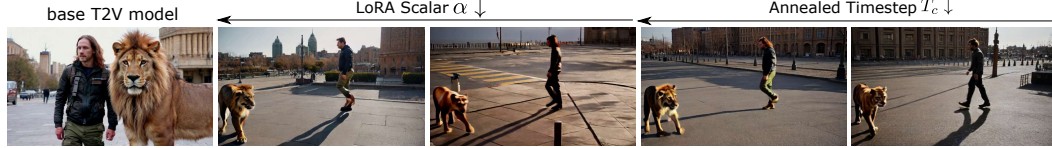

Figure 7: **Ablation Results on Domain Adaptor (upper) and Annealed Sampling (the bottom).** We provide more experiments in Sec. F.2.1 to choose suitable $\alpha$ and $T_c$ to improve video quality.

**Improving Video Quality.** As illustrated in Fig. 7 and Table 3, without the video domain adaptor, the video quality deteriorates significantly, reverting to a purely UE-style appearance similar to the training set. Likewise, omitting the annealed sampling strategy results in a decline in video quality (see the beard of the lion and overall scene style). While the rotation accuracy drops slightly (0.277→0.265), this is acceptable since there exist errors in evaluating open-world human poses.

**Motion Fusion Design.** As shown in Table 3, replacing gated self-attention with cross-attention fusion (w/ Cross-Attn. Fusion, here we use the entity-motion bonded feature $\mathbf{Z}^{\mathbf{Pe}}$ as the query) or placing the object injector after the 3D self-attention layer (w/ 3D Self-Attn.) results in a slight decline in both video quality and pose sequence accuracy.

## 5 CONCLUSION

In this work, we introduce 3DTrajMaster, a unified framework for controlling multi-entity motions in 3D space, with motion representation as 6DoF location and rotation sequences. Our flexible object injector establishes entity-wise correspondence and allows flexible editing of entity descriptions.

**Limitation.** Generalizable entities, like animals, cannot be edited with the same level of granularity as humans. This limitation can be addressed by constructing more diverse and detailed 3D assets of the same category. Currently, the model is constrained to global motion patterns; however, fine-grained local motions (e.g., human dancing or waving hands) and interactions between different entities (e.g., a man picking up a dog) can also be modeled similarly to our 6 DoF motions with structured motion patterns. At present, our model can only generate limited entities (≤3) at a time, but this can be improved with more powerful video foundation models and paired datasets.

ACKNOWLEDGMENTS

We thank Jinwen Cao, Yisong Guo, Haowen Ji, Jichao Wang, and Yi Wang from Kuaishou Technology for their help in constructing our 360°-Motion Dataset. As for the fruitful discussion, we thank Yuzhou Huang, Qinghe Wang, Runsen Xu, Zeqi Xiao, and Zhouxia Wang.

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
