APPENDIX

## A   INTERNAL VIDEO DIFFUSION MODEL FOR RESEARCH PURPOSE

Our model is a transformer-based latent diffusion model, as illustrated in the Fig. S8. Initially, we employ a 3D VAE to transform videos from the pixel level to a latent space, upon which we construct a transformer-based video diffusion model (Peebles & Xie, 2023). Previous models, which rely on UNets (Blattmann et al., 2023; Chen et al., 2023; Guo et al., 2023b) or transformers (Ma et al., 2024), typically incorporate an additional 1D temporal attention module for video generation, and such spatial-temporally separated designs do not yield optimal results. Instead, we replace the 1D temporal attention with 3D self-attention (Gupta et al., 2023), enabling the model to more effectively perceive and process spatiotemporal tokens, thereby achieving a high-quality and coherent video generation model. Specifically, we map the timestep to a scale, thereby applying RMSNorm to the spatiotemporal tokens before each attention or feed-forward network (FFN) module.

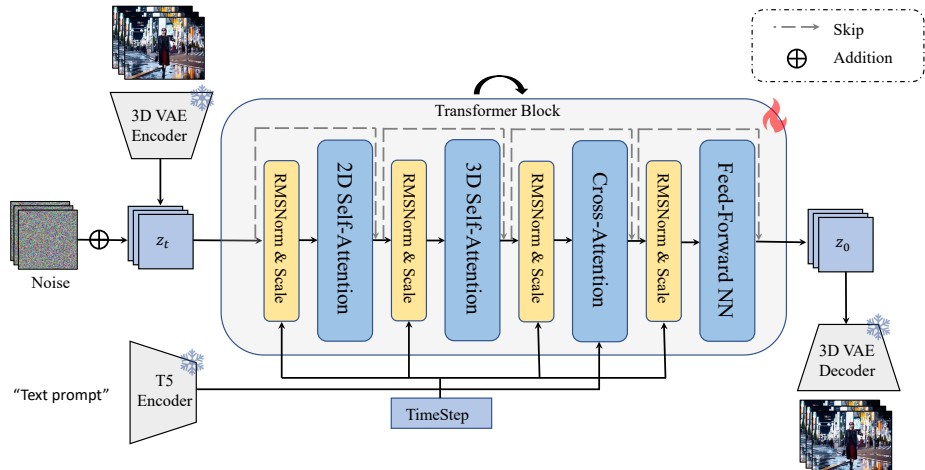

Figure S8: **Our Video Latent Diffusion Model Backbone**

## B   ADDITIONAL RELATED WORK

**Injecting Control into Video Foundation Models.** *(1) Learning-based*: The control signals are typically projected into latent embeddings via an extra encoder (e.g., learnable convolutional/linear/attention/LoRA layers, or frozen pre-trained feature encoder), which are then integrated into the base model architecture through concatenation, addition, or insertion. VideoComposer[1] employs a unified STC-encoder and CLIP model to feed multi-modal input conditions (textual, spatial, and temporal) into the base T2V model. MotionCtrl (Wang et al., 2024c) introduces camera motion by fine-tuning specific layers of the base U-Net, and object motion via additional convolutional layers. CameraCtrl (He et al., 2024) enhances this approach by incorporating ControlNet (Zhang et al., 2023)'s philosophy, using an attention-based pose encoder to fuse camera signals in the form of Plücker embeddings while keeping the base model frozen. Similarly, SparseCtrl (Guo et al., 2023a) learns an add-on encoder to integrate control signals (RGB, sketch, depth) into the base model. Tora (Zhang et al., 2024) employs a trajectory encoder and plug-and-play motion fuser to merge 2D trajectories with the base video model. MotionDirector[7] leverages spatial and temporal LoRA layers to learn desired motion patterns from reference videos. *(2) Training-free*: These methods modify attention layers or video latents to adjust control signals in a computationally efficient manner. However, training-free methods often suffer from poor generalization and require extensive trial-and-error. Direct-a-video (Yang et al., 2024) amplifies or suppresses attention in spatial cross-attention layers to inject box guidance, while FreeTraj (Qiu et al., 2024) embeds target trajectories into the low-frequency components and redesigns reweighting strategies across attention layers. MOFT (Xiao et al., 2024) extracts motion priors by removing content correlation and applying motion channel filtering, and then alters the sampling process using the reference MOFT.

## C ADDITIONAL APPLICATIONS

We outline our potential applications in various areas as follows.

1) **Film:** Reproduce the character's classic moves. We can extract the human poses from a given video and apply them to different entities and backgrounds using the capabilities of our model.

2) **Autonomous Driving:** Simulate dangerous safety accidents, such as two cars colliding and a car hitting a person.

3) **Embodied AI:** Generate a vast number of videos with diverse entity and trajectory inputs to train a general 4D pose estimator, especially for non-rigid objects.

4) **Game:** Train a character ID, such as Black Myth Wukong, through LoRA, and then drive the character movement with different trajectories.

## D CLARIFICATION OF THE LIMITED ENTITY NUMBER ($\leq 3$)

Currently, our method is limited to generating up to 3 entities, as outlined in the 'Limitation' section of the paper. This constraint is primarily due to the capabilities of the video foundation model rather than the training data. While it is relatively easy to generate $\gg 2$ entities of the same category (e.g., "a group of people/cars/animals") in the video, it becomes much more challenging to generate $\gg 2$ entities, each differing greatly from the others, through the text input as T5 text encoder tends to mix the textual features of different entities. Thus it becomes hard to associate specific trajectories with their corresponding text entities. Based on empirical studies with video foundation models, we chose to limit the number of entities to 3 in our work. Regarding data construction, it is easy to include more entities with their paired trajectories in our procedure UE platform pipeline. However, the key limitation is that the video foundation model struggles to generate such a diverse set of entities simultaneously. Furthermore, many prior works, such as Tora, MotionCtrl, and Direct-a-video also focus on a limited number of entities.

## E DATASET ILLUSTRATION

### E.1 360°-MOTION DATASET DATA.

We show a sample in Fig. S9 captured with 12 evenly-surrounded cameras. Each camera shoots a clip of 100 frames at $384 \times 672$ resolutions. During training, we discard the initial 10 frames to eliminate potential blurring and noise caused by 3D model initialization in the UE platform.

### E.2 UNBALANCED ENTITY DISTRIBUTION IN COMMON VIDEO DATASETS

In high-quality video datasets like Artgrid, Pixabay, and Pexels[3], the issue of category imbalance is highly pronounced and poses significant challenges. We analyze the aforementioned three datasets by first captioning the videos using QWen-VL (Bai et al., 2023). Subsequently, we employ the spaCy[4] library to extract noun chunks from the video captions, which serve as entity words. We predefine over 60 classes as keywords for entity filtering. As illustrated in the Fig. S10, certain categories (e.g., humans) constitute a disproportionately large share of the entity objects, thereby constraining the model's ability to generalize to other categories that appear less frequently.

### E.3 GPT-GENERATED EVALUATION PROMPTS

The human prompts, non-human (animal, car, robot) prompts, and location prompts for evaluation are provided in Table R4, Table R5&Table R6, and Table R7 respectively.

---

[3] Artgrid: https://artgrid.io/, Pixabay: https://www.videvo.net/, Pexels: https://www.pexels.com/
[4] spaCy: https://spacy.io/

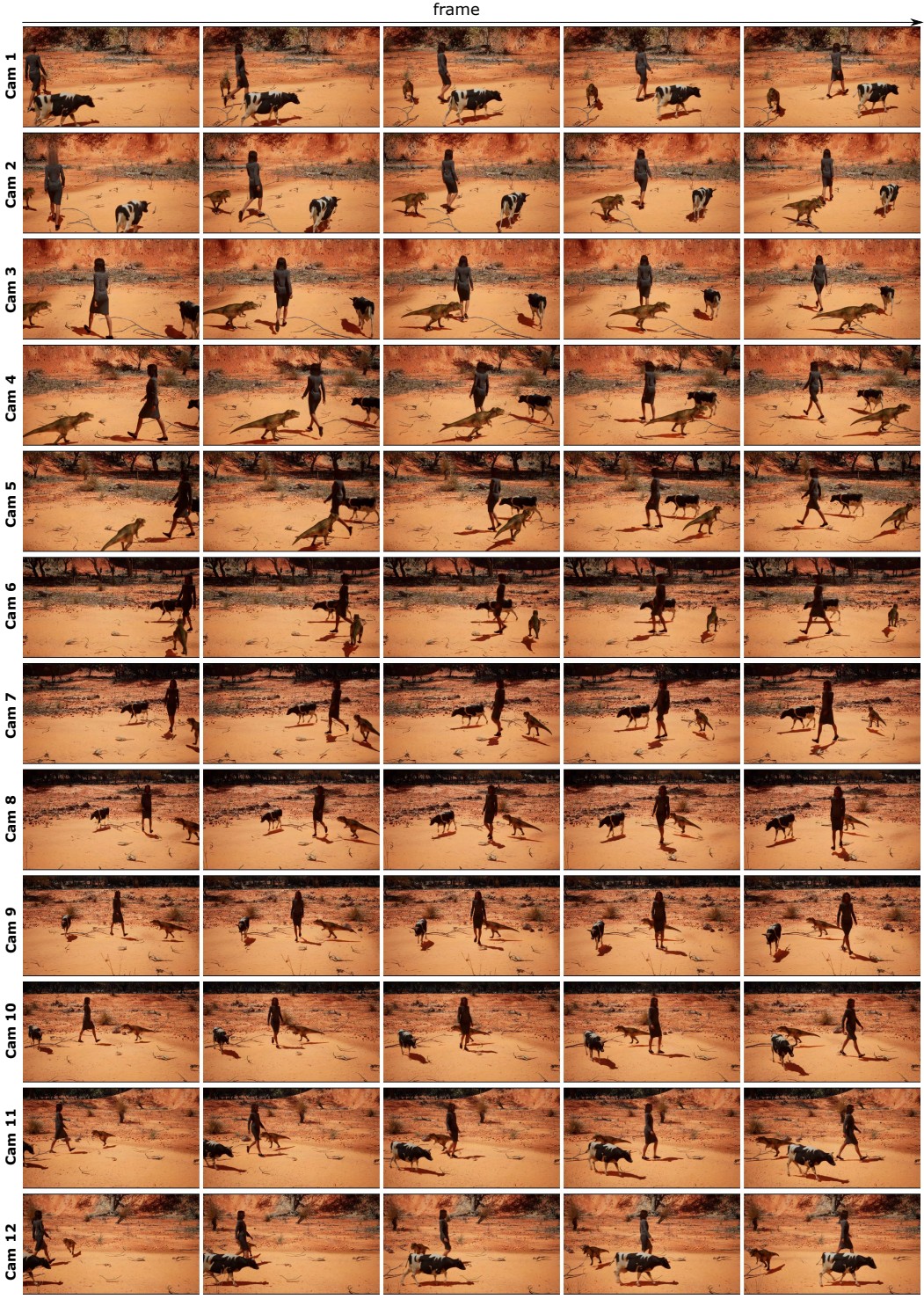

Figure S9: **A Sample from our *360°-Motion* Captured with 12 Evenly-Surrounded Cameras.**

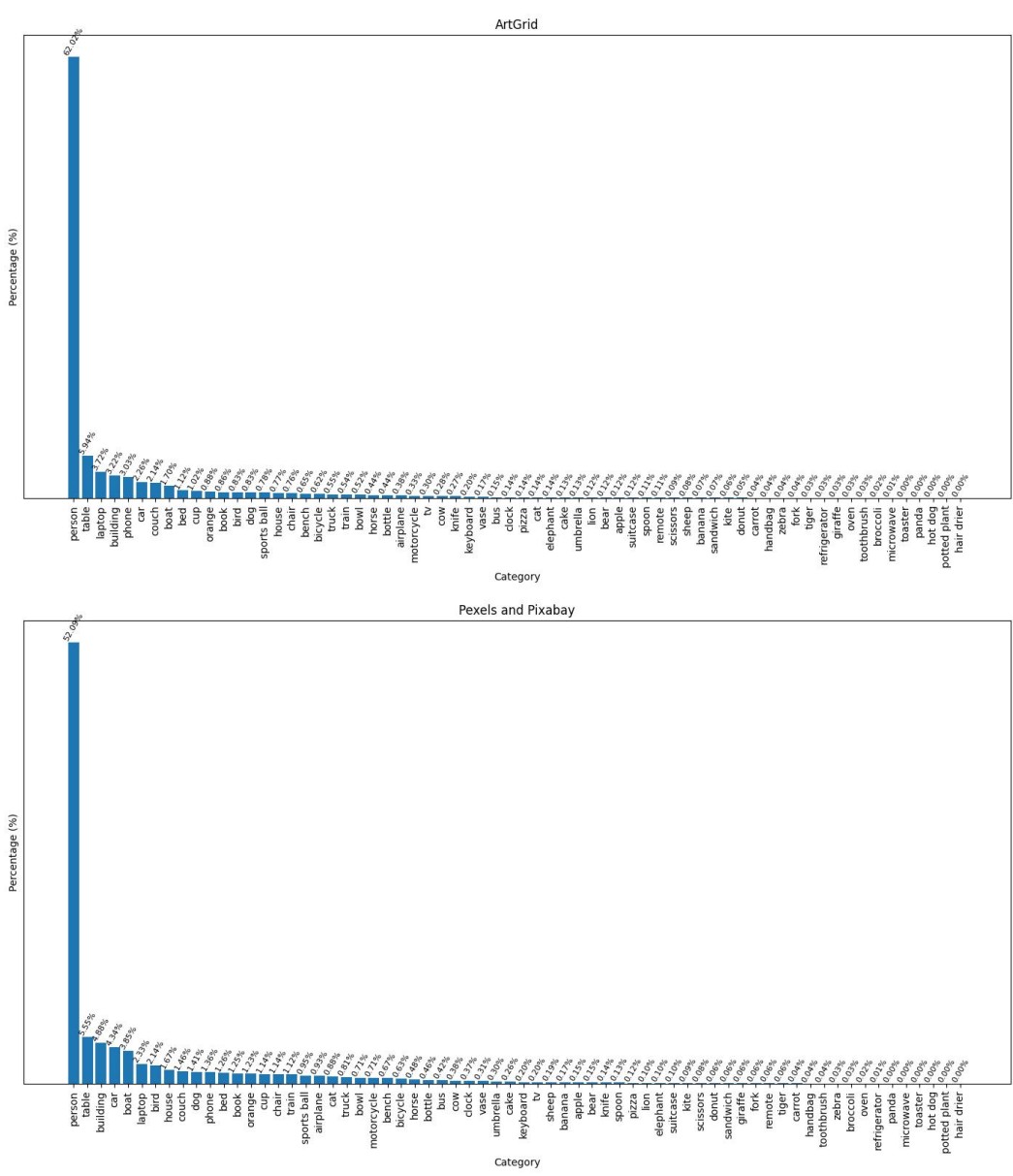

Figure S10: **Entity Distribution Over 60 Classes in Artgrid, Pixabay, and Pexels.**

# F   MORE EXPERIMENTS

## F.1   FINE-GRAINED ENTITY PROMPT INPUT

We provide additional samples in Fig. S11 to demonstrate that 3DTrajMaster supports fine-grained entity customization. The description of the man can be flexibly modified by adjusting attributes such as hair, gender, physique, clothing, and accessories.

Table R4: **Evaluation Human Prompts**. They are generated using GPT prompt: *"Generate more human samples similar to {Train Human Sample}, no more than 25 words."*

1. a man with short spiky brown hair, athletic build, a navy blue jacket, beige cargo pants, and black sneakers
2. a woman with long wavy blonde hair, petite figure, a red floral dress, white sandals, and a yellow shoulder bag
3. a man with a shaved head, broad shoulders, a gray graphic t-shirt, dark jeans, and brown leather boots
4. a woman with shoulder-length straight auburn hair, a slender figure, a green button-up blouse, black leggings, and white sneakers
5. a man with messy black hair, tall frame, a plaid red and black shirt, faded blue jeans, and tan hiking boots
6. a man with medium-length straight brown hair, tall and slender, a gray crew-neck t-shirt, beige trousers, and dark green sneakers
7. a woman with short curly black hair, slender build, a pink hoodie, light gray joggers, and blue sneakers
8. a man with short black wavy hair, lean figure, a green and yellow plaid shirt, dark brown pants, and black suede shoes
9. a man with curly black hair, muscular build, a dark green hoodie, gray joggers, and white running shoes
10. a woman with short blonde hair, slim athletic build, a red leather jacket, dark blue jeans, and white sneakers
11. a man with medium-length wavy brown hair, lean build, a black bomber jacket, olive green cargo pants, and brown hiking boots
12. a man with buzz-cut blonde hair, stocky build, a gray zip-up sweater, black shorts, and red basketball shoes
13. a woman with long straight black hair, toned build, a blue denim jacket, light gray legg -ings, and black slip-on shoes
14. a man with short curly red hair, average build, a black leather jacket, dark blue cargo pants, and white sneakers
15. a woman with shoulder-length wavy brown hair, slim build, a green parka, black leggings, and gray hiking boots
16. a man with short straight black hair, tall and lean build, a navy blue sweater, khaki shorts, and brown sandals
17. a woman with pixie-cut blonde hair, athletic build, a red windbreaker, blue ripped jeans, and black combat boots
18. a man with medium-length wavy gray hair, muscular build, a maroon t-shirt, beige chinos, and brown loafers
19. a woman with long curly black hair, average build, a purple hoodie, black athletic shorts, and white running shoes
20. a man with short spiky blonde hair, slim build, a black trench coat, blue jeans, and brown hiking shoes

## F.2 ABLATION STUDY

### F.2.1 OPTIMAL HYPERPARAMETERS

In the main paper, we propose a video domain adaptor and an annealed sampling strategy to mitigate video domain shifts from our constructed UE datasets. However, completely removing the LoRA adaptor (as the learned motion and domain bias are coupled to some extent) or the inserted motion guidance will result in a decline in 3D trajectory accuracy. Thus, applying video enhancement techniques with appropriate dropping is crucial. To this end, we begin with a randomly initialized parameter group: $T_c = 10, \alpha = 0.2, TS = 72,000$. We perform ablation experiments on our evaluation subset. As shown in Table R8, Table R9, and Table R10, the video quality exhibits a monotonically decreasing trend as these hyperparameters increase. In contrast, 3D trajectory accuracy initially drops sharply but stabilizes in the later stages. To balance the degradation of visual quality with

Table R5: **Evaluation Non-Human Prompts (1/2)**. They are generated using GPT prompt: *"Generate more animal/car/robot samples similar to {Train Sample}, no more than 25 words."*

1. a dog with a fluffy coat, wagging tail, and warm golden-brown fur, exuding a gentle and friendly charm
2. a tiger with vibrant orange and black stripes, piercing yellow eyes, and a powerful stance, exuding strength and grace
3. a giraffe with golden-yellow fur, long legs, a tall slender neck, and patches of brown spots, exuding elegance and calm
4. an alpaca with soft white wool, short legs, a thick neck, and a fluffy head of fur, radiating gentle charm
5. a zebra with black and white stripes, sturdy legs, a short neck, and a sleek mane running down its back
6. a deer with sleek tan fur, long slender legs, a graceful neck, and tiny antlers atop its head
7. a gazelle with light golden fur, long slender legs, a thin neck, and short, sharp horns, embodying elegance and agility
8. a horse with chestnut brown fur, muscular legs, a slim neck, and a flowing mane, exuding strength and grace
9. a sleek black panther with a smooth, glossy coat, emerald green eyes, and a powerful stance
10. a cheetah with golden fur covered in black spots, intense amber eyes, and a slender, agile body
11. a regal lion with a thick, flowing golden mane, sharp brown eyes, and a powerful muscular frame
12. a snow leopard with pale gray fur adorned with dark rosettes, icy blue eyes, and a stealthy, poised posture
13. a jaguar with a golden-yellow coat dotted with intricate black rosettes, deep green eyes, and a muscular build
14. a wolf with thick silver-gray fur, alert golden eyes, and a lean yet strong body, exuding confidence and boldness
15. a tiger with a pristine white coat marked by bold black stripes, bright blue eyes, and a graceful, poised form
16. a lynx with tufted ears, soft reddish-brown fur with faint spots, and intense yellow-green eyes
17. a bear with dark brown fur, small but fierce black eyes, and a broad and muscular build, radiating power
18. a swift fox with reddish-orange fur, a bushy tail tipped with white, and sharp, intelligent amber eyes
19. a falcon with blue-gray feathers, sharp talons, and keen yellow eyes fixed on its prey below
20. a fox with sleek russet fur, a bushy tail tipped with black, and bright green and cunning eyes
21. a kangaroo with brown fur, powerful hind legs, and a muscular tail, showcasing its strength and agility
22. a polar bear with thick white fur, strong paws, and a black nose, embodying the essence of the Arctic
23. a cheetah with a slender build, spotted golden fur, and sharp eyes, epitomizing speed and agility
24. a dolphin with sleek grey skin, a curved dorsal fin, and intelligent, playful eyes, reflecting its nature
25. a wolf with a body covered in thick silver fur, sharp ears, and piercing yellow eyes, showcasing its alertness
26. a leopard with a body covered in golden fur, dark rosettes, and a long muscular tail, emphasizing its strength
27. a penguin with a body covered in smooth black-and-white feathers, short wings, and webbed feet
28. a gazelle with a body covered in sleek tan fur, long legs, and elegant curved horns, showcasing its grace

maintaining pose accuracy, we select an optimal parameter group: $T_c = 25, \alpha = 0.4, TS = 36,000$ as our default inference setting.

Table R6: **Evaluation Non-Human Prompts (2/2)**. They are generated using GPT prompt: *"Generate more animal/car/robot samples similar to {Train Sample}, no more than 25 words."*

29. a rabbit with a body covered in soft fur, quick hops, and a playful demeanor, showcasing its energy
30. a koala with a body covered in soft grey fur, large round ears, and a black nose, radiating cuteness
31. a rhinoceros with a body covered in thick grey skin, a massive horn on its snout, and sturdy legs
32. a flamingo with a body covered in pink feathers, long slender legs, and a gracefully curved neck
33. a parrot with bright red, blue, and yellow feathers, a curved beak, and sharp eyes
34. a hippopotamus with a body covered in thick grey-brown skin, massive jaws, and a large body
35. a crocodile with a body covered in scaly green skin, a powerful tail, and sharp teeth
36. a moose with a body covered in thick brown fur, massive antlers, and a bulky frame
37. a fluttering butterfly with intricate wing patterns, vivid colors, and graceful flight
38. a chameleon with a body covered in vibrant green scales, bulging eyes, and a curled tail, showcasing its unique charm
39. a lemur with a body covered in soft grey fur, a ringed tail, and wide yellow eyes, and curious expression
40. a squirrel with a body covered in bushy red fur, large eyes, and a fluffy tail
41. a panda with a body covered in fluffy black-and-white fur, a round face, and gentle eyes, radiating warmth
42. a porcupine with a body covered in spiky brown quills, a small nose, and curious eyes
43. a sedan with a sleek metallic silver body, long wheelbase, a low-profile hood, and a small rear spoiler
44. an SUV with a matte black exterior, elevated suspension, a tall roofline, and a compact rear roof rack
45. a pickup truck with rugged dark green paint, extended cab, raised suspension, and a modest cargo bed cover
46. a vintage convertible with a body covered in shiny red paint, chrome bumpers, and a stylish design
47. a futuristic electric car with a minimalist silver design, slim LED lights, and smooth curves
48. a compact electric vehicle with a silver finish, aerodynamic profile, and efficient battery
49. a firefighting robot with a water cannon arm, heat sensors, and durable red-and-silver exterior
50. an industrial welding robot with articulated arms, a laser precision welder, and heat-resistant shields
51. a disaster rescue robot with reinforced limbs, advanced AI, and a rugged body designed to navigate
52. an exploration rover robot with solar panels, durable wheels, and advanced sensors for planetary exploration

Table R7: **Evaluation Location Prompts**.

1. fjord 2. sunset beach 3. cave 4. snowy tundra 5. prairie 6. asian town 7. rainforest 8. canyon
9. savanna 10. urban rooftop garden 11. swamp 12. riverbank 13. coral reef 14. volcanic landscape
15. wind farm 16. town street 17. night city square 18. mall lobby 19. glacier 20. seaside street
21. gymnastics room 22. abandoned factory 23. autumn forest 24. mountain village 25. coastal harbor
26. ancient ruins 27. modern metropolis 28. dessert 29. forest 30. city 31. snowy street 32. park

### F.2.2 NEGATIVE POSE CONDITION AS STATIC MOTIONS

We find that setting negative pose sequences as static motions $\{(\hat{\mathbf{P}}_n)_{n=1}^N | \hat{\mathbf{P}}_n = \mathbf{P}_0, \forall n\}$ rather than positive motion sequences $\{(\mathbf{P}_n)_{n=1}^N\}$ can further improve pose accuracy, as shown in Table R11. We infer that the model captures underlying 3D motion representations from the randomly generated 3D trajectories. However, we do not adopt this approach due to the decline in video quality.

frame

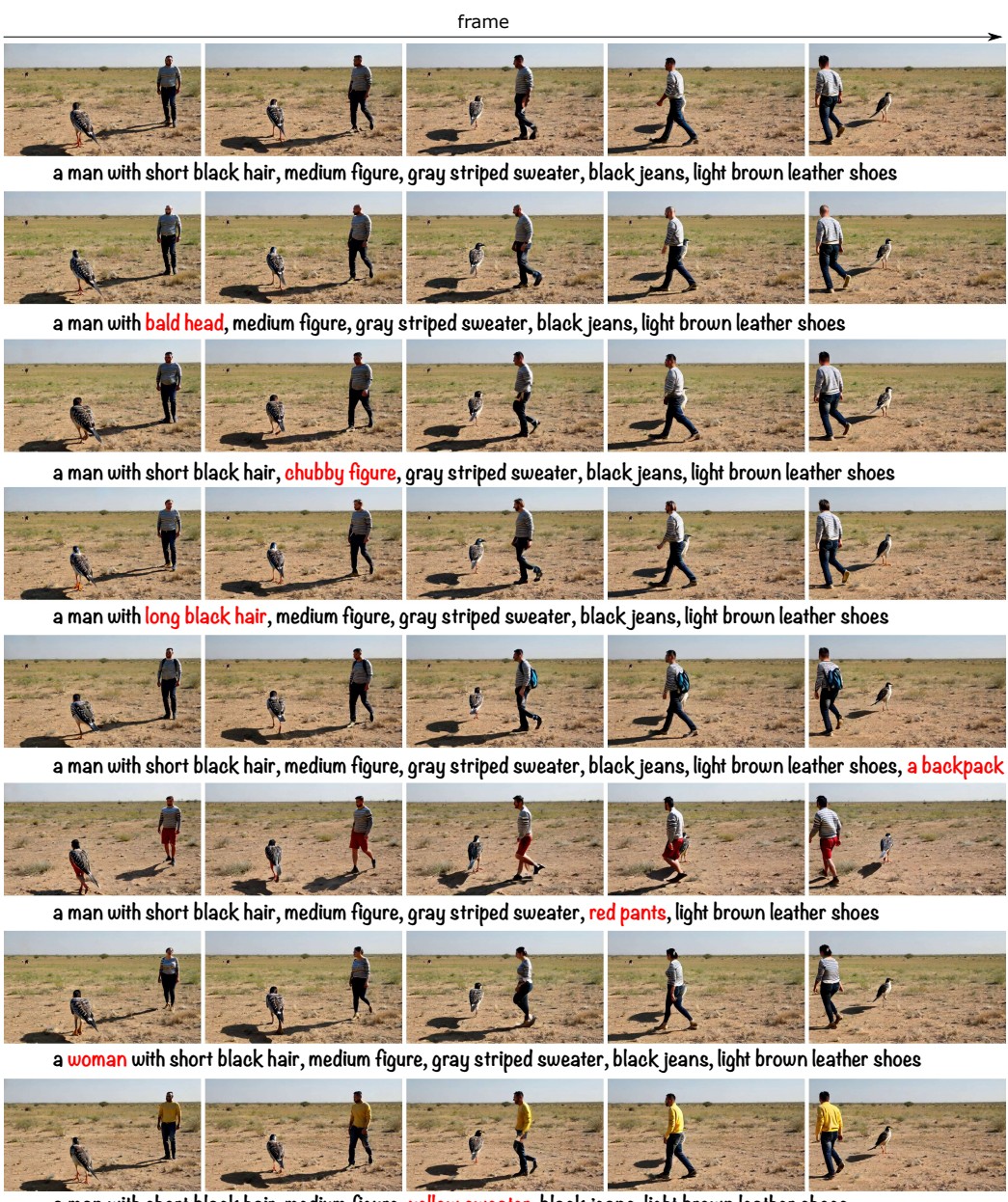

Figure S11: **Flexible Entity Editing in Input Text Prompts.** The other entity, *"a swift falcon with blue-gray feathers, sharp talons, and keen yellow eyes focused on its prey below"* remains fixed while varying the human entity descriptions.

### F.2.3 QUALITATIVE FEEDBACK FROM HUMAN USERS

We conducted a questionnaire survey and collected 53 samples to form user preference comparisons. Each participant received a reward of 0.80 USD and spent approximately 5 minutes completing the questionnaire, which assessed four dimensions: (1) video quality, (2) trajectory accuracy, (3) entity diversity, and (4) background diversity. In Table R12, we report the proportion of users who preferred our model over the baselines.

Table R8: Ablation Study on Annealed Timestep $T_c$.

| Annealed Timestep $T_c$ | Video Quality | | | 3D Trajectory Accuracy | |
|---|---|---|---|---|---|
| | FVD ↓ | FID ↓ | CLIPSIM ↑ | TransErr (m) ↓ | RotErr (deg) ↓ |
| $T_c = 5$ | **1492.79** | **76.95** | **0.3469** | 0.844 | 1.099 |
| $T_c = 10$ | 1976.01 | 106.45 | 0.3429 | 0.546 | 0.493 |
| $T_c = 15$ | 2179.15 | 122.55 | 0.3405 | 0.437 | 0.422 |
| $T_c = 20$ | 2236.05 | 128.89 | 0.3374 | 0.391 | 0.284 |
| $T_c = 25$ | 2240.40 | 132.90 | 0.3337 | **0.344** | 0.274 |
| $T_c = 30$ | 2295.13 | 137.52 | 0.3314 | 0.360 | **0.261** |
| $T_c = 35$ | 2323.20 | 142.71 | 0.3276 | 0.352 | 0.264 |
| $T_c = 40$ | 2338.47 | 148.27 | 0.3240 | 0.351 | 0.266 |
| $T_c = 45$ | 2363.49 | 156.39 | 0.3207 | 0.350 | 0.268 |
| $T_c = 50$ | 2347.64 | 166.71 | 0.3185 | 0.348 | 0.281 |

Table R9: Ablation Study on LoRA Scalar $\alpha$.

| LoRA Scalar $\alpha$ | Video Quality | | | 3D Trajectory Accuracy | |
|---|---|---|---|---|---|
| | FVD ↓ | FID ↓ | CLIPSIM ↑ | TransErr (m) ↓ | RotErr (deg) ↓ |
| $\alpha = 0$ | **1495.38** | **80.56** | **0.3467** | 0.646 | 0.900 |
| $\alpha = 0.2$ | 1976.01 | 106.45 | 0.3429 | 0.546 | 0.493 |
| $\alpha = 0.4$ | 2150.42 | 133.76 | 0.3367 | 0.444 | 0.428 |
| $\alpha = 0.6$ | 2330.56 | 152.12 | 0.3277 | 0.394 | **0.393** |
| $\alpha = 0.8$ | 2318.78 | 195.93 | 0.3125 | 0.378 | 0.450 |
| $\alpha = 1.0$ | 2481.33 | 224.81 | 0.3087 | **0.358** | 0.432 |

Table R10: Ablation Study on Training Step $TS$.

| Train. Steps $TS$ | Video Quality | | | 3D Trajectory Accuracy | |
|---|---|---|---|---|---|
| | FVD ↓ | FID ↓ | CLIPSIM ↑ | TransErr (m) ↓ | RotErr (deg) ↓ |
| $TS = 12,000$ | **1493.68** | **72.03** | 0.3427 | 0.561 | 0.713 |
| $TS = 36,000$ | 1883.15 | 99.98 | 0.3408 | 0.523 | 0.631 |
| $TS = 72,000$ | 1976.01 | 106.45 | **0.3429** | 0.546 | 0.493 |
| $TS = 108,000$ | 2068.43 | 111.01 | 0.3388 | 0.446 | **0.480** |
| $TS = 144,000$ | 2102.28 | 114.84 | 0.3367 | **0.411** | 0.482 |

Table R11: Ablation Study on Negative Pose Sequences.

| Negative Condition | Video Quality | | | 3D Trajectory Accuracy | |
|---|---|---|---|---|---|
| | FVD ↓ | FID ↓ | CLIPSIM ↑ | TransErr (m) ↓ | RotErr (deg) ↓ |
| Neg. Pose = Static Motions | 2141.39 | 118.22 | 0.3360 | **0.371** | **0.448** |
| Neg. Pose = Pos. Pose | **1976.01** | **106.45** | **0.3429** | 0.546 | 0.493 |

Table R12: User Preference Comparisons.

| Method | MotionCtrl | Direct-a-Video | Tora |
|---|---|---|---|
| 3DTrajMaster | 47.2% | 56.6% | 81.1% |

### F.2.4 GENERALIZABLE ENTITY PROMPTS&3D TRAJECTORIES

We provide more generalizable results with novel entity prompts generated by GPT and 3D trajectories, as shown in Fig. S12 to Fig. S31. Each text prompt consists of one to three entities. *(We kindly urge readers to check the visual results in the our website)*.

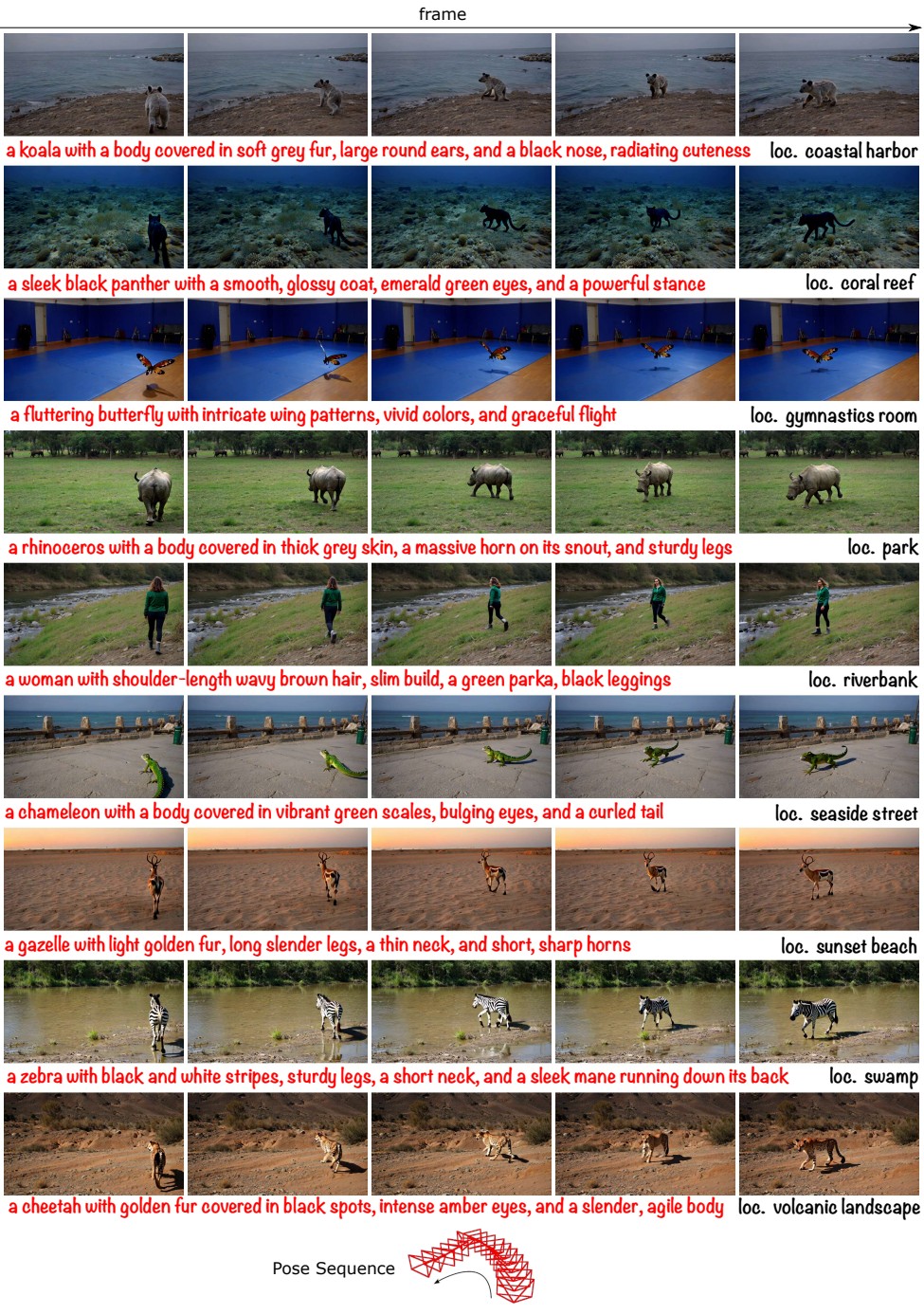

Figure S12: **Generalizable Results with Novel 3D Trajectories & Entity Prompts (1/20)**

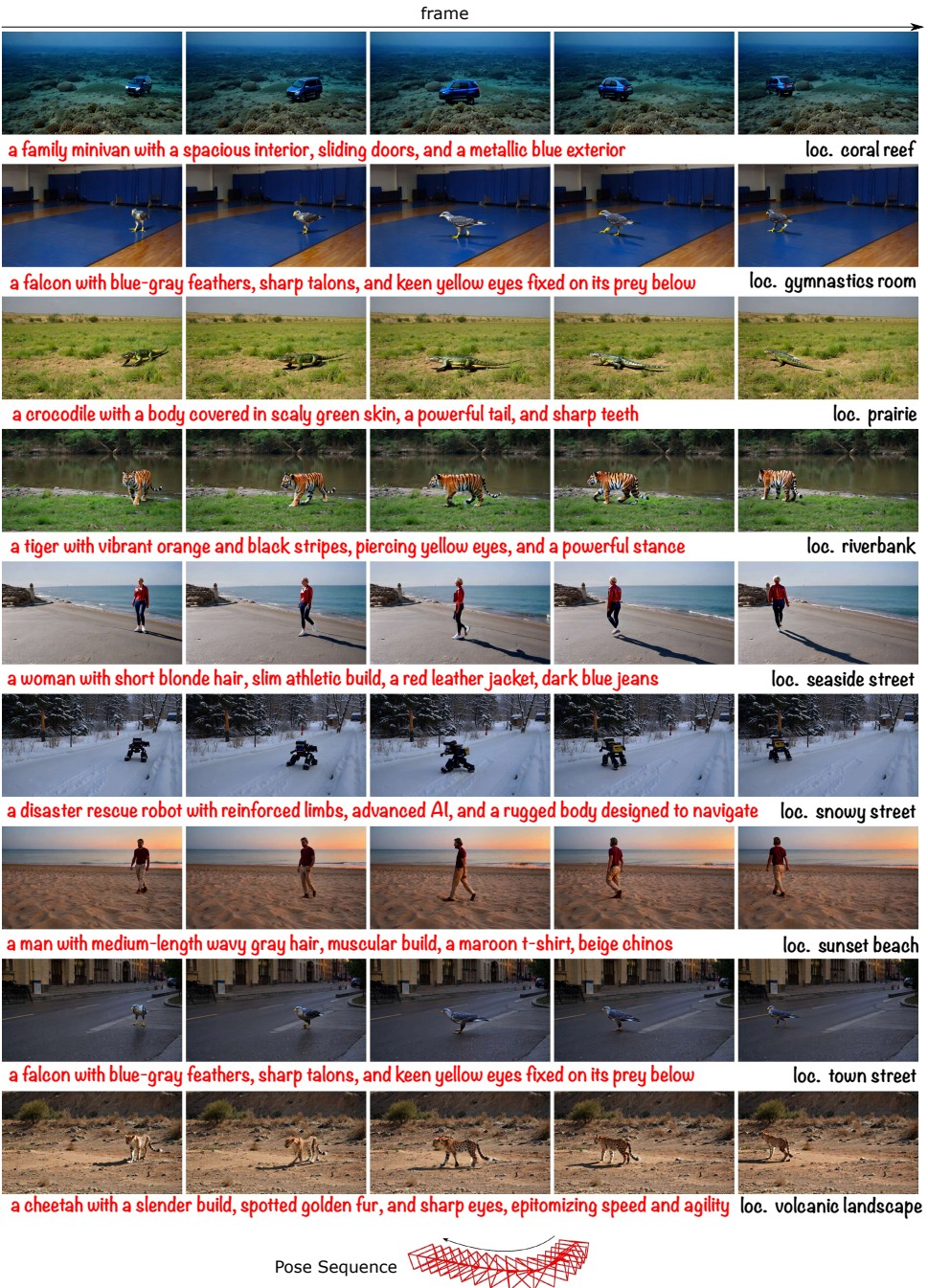

Figure S13: **Generalizable Results with Novel 3D Trajectories & Entity Prompts (2/20)**

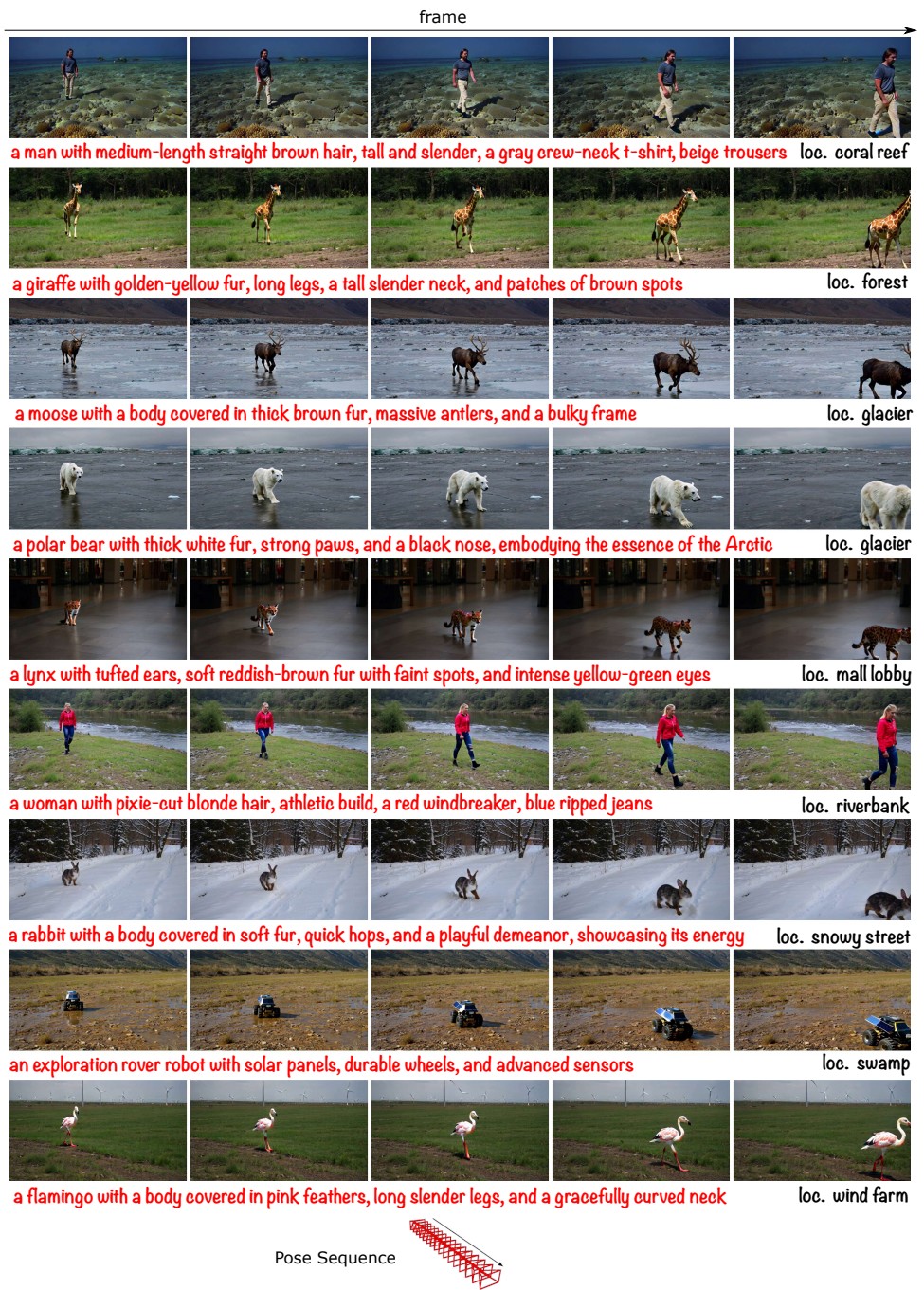

Figure S14: **Generalizable Results with Novel 3D Trajectories & Entity Prompts (3/20)**

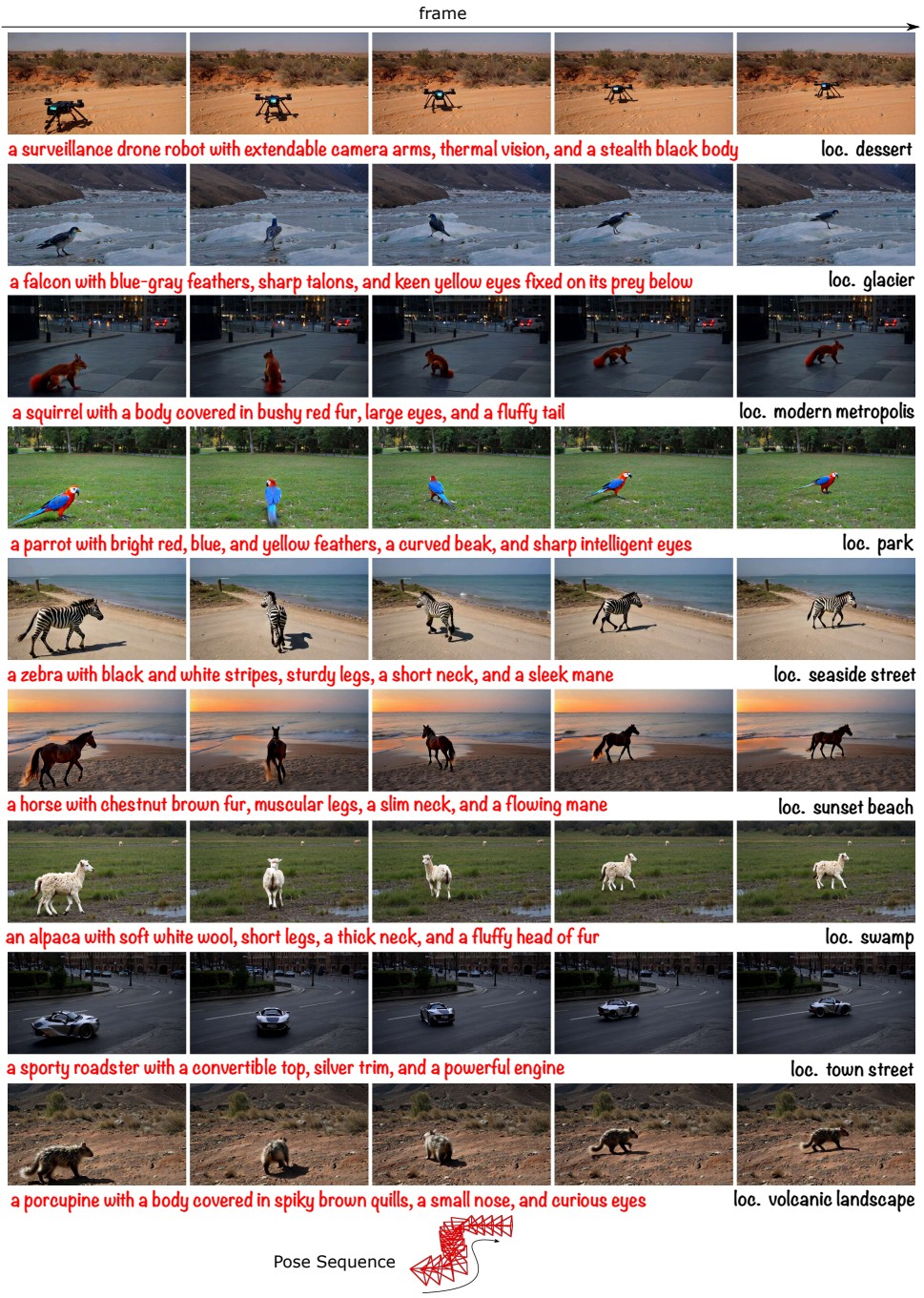

Figure S15: **Generalizable Results with Novel 3D Trajectories & Entity Prompts (4/20)**

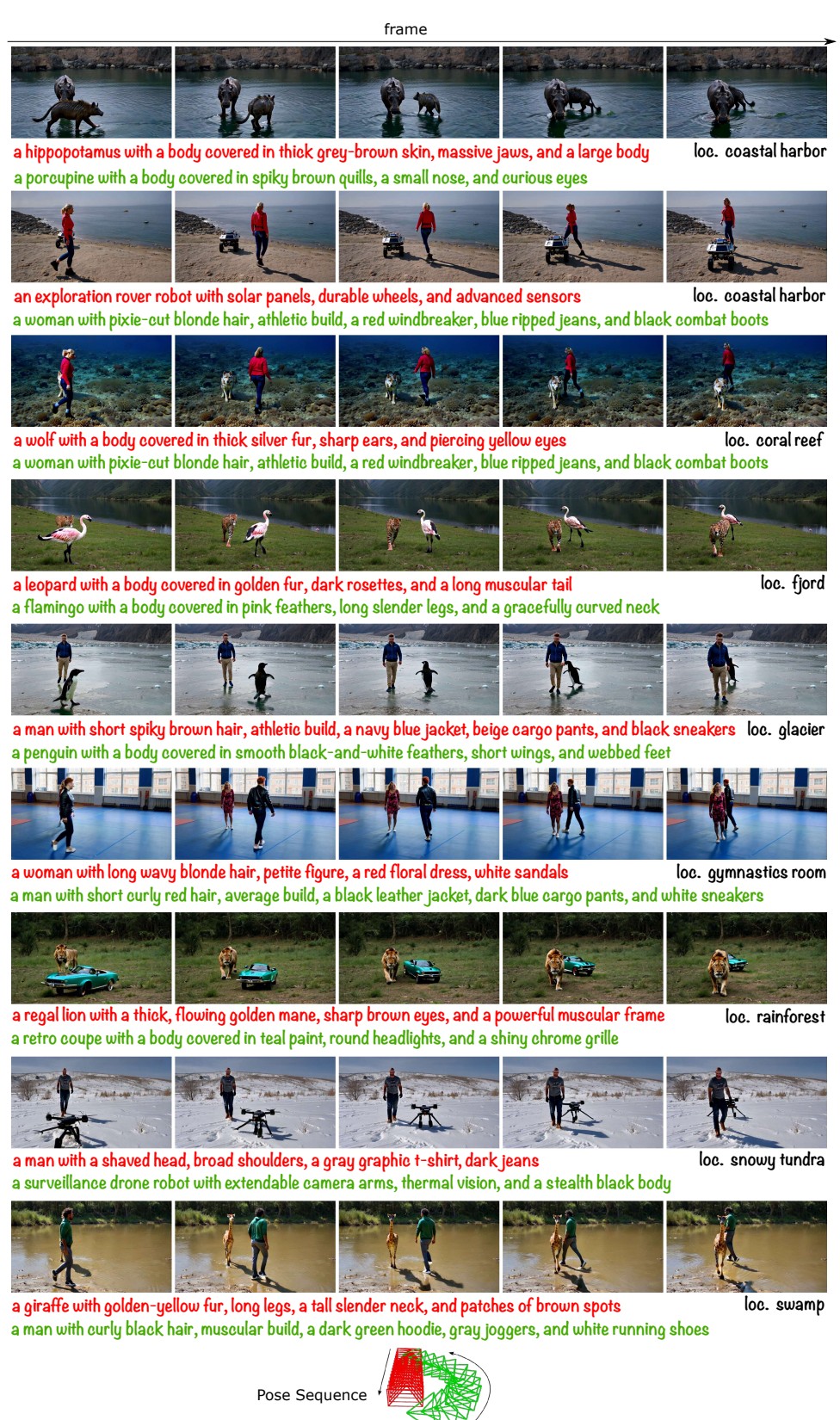

Figure S16: **Generalizable Results with Novel 3D Trajectories & Entity Prompts (5/20)**

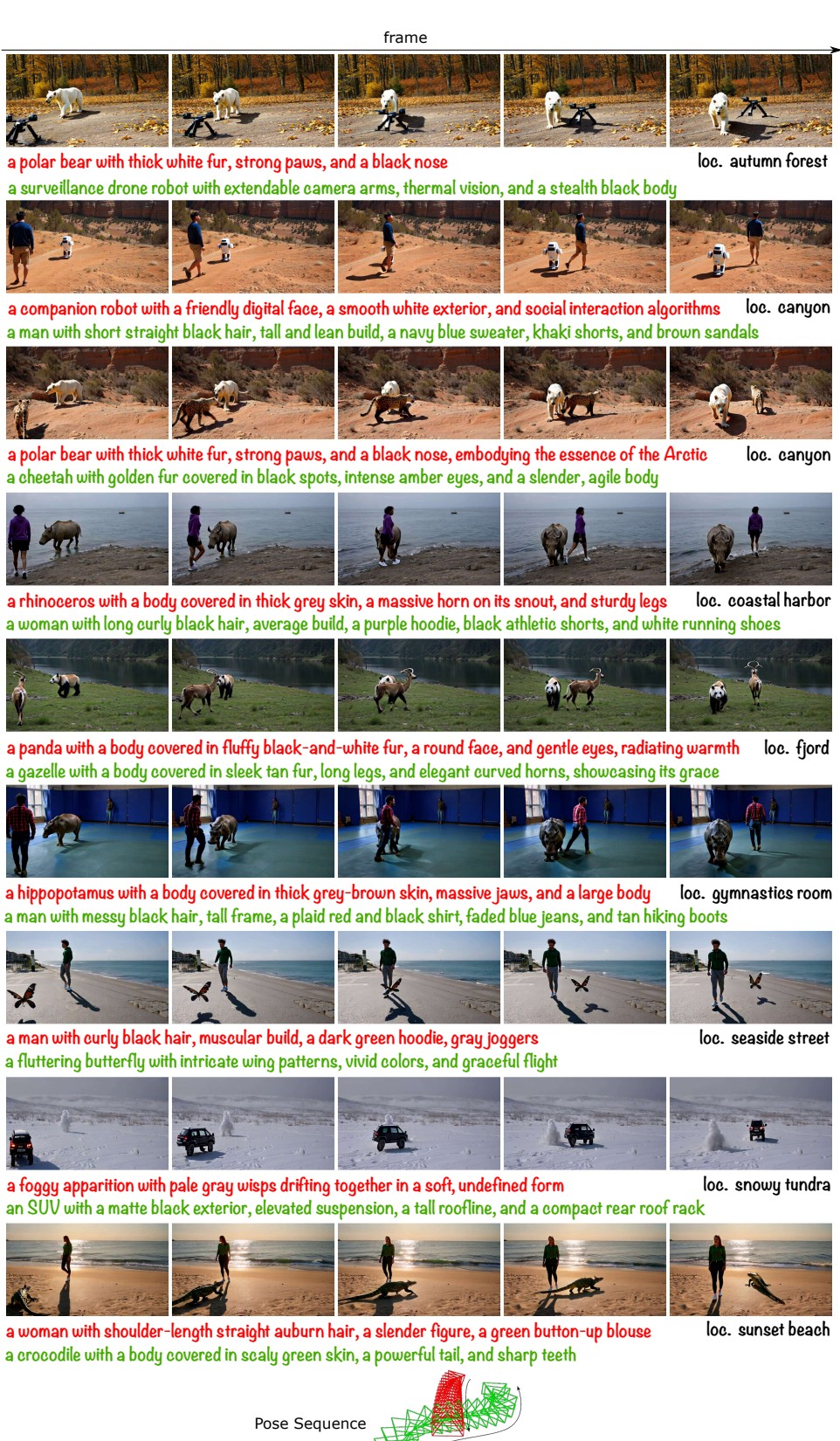

Figure S17: **Generalizable Results with Novel 3D Trajectories & Entity Prompts (6/20)**

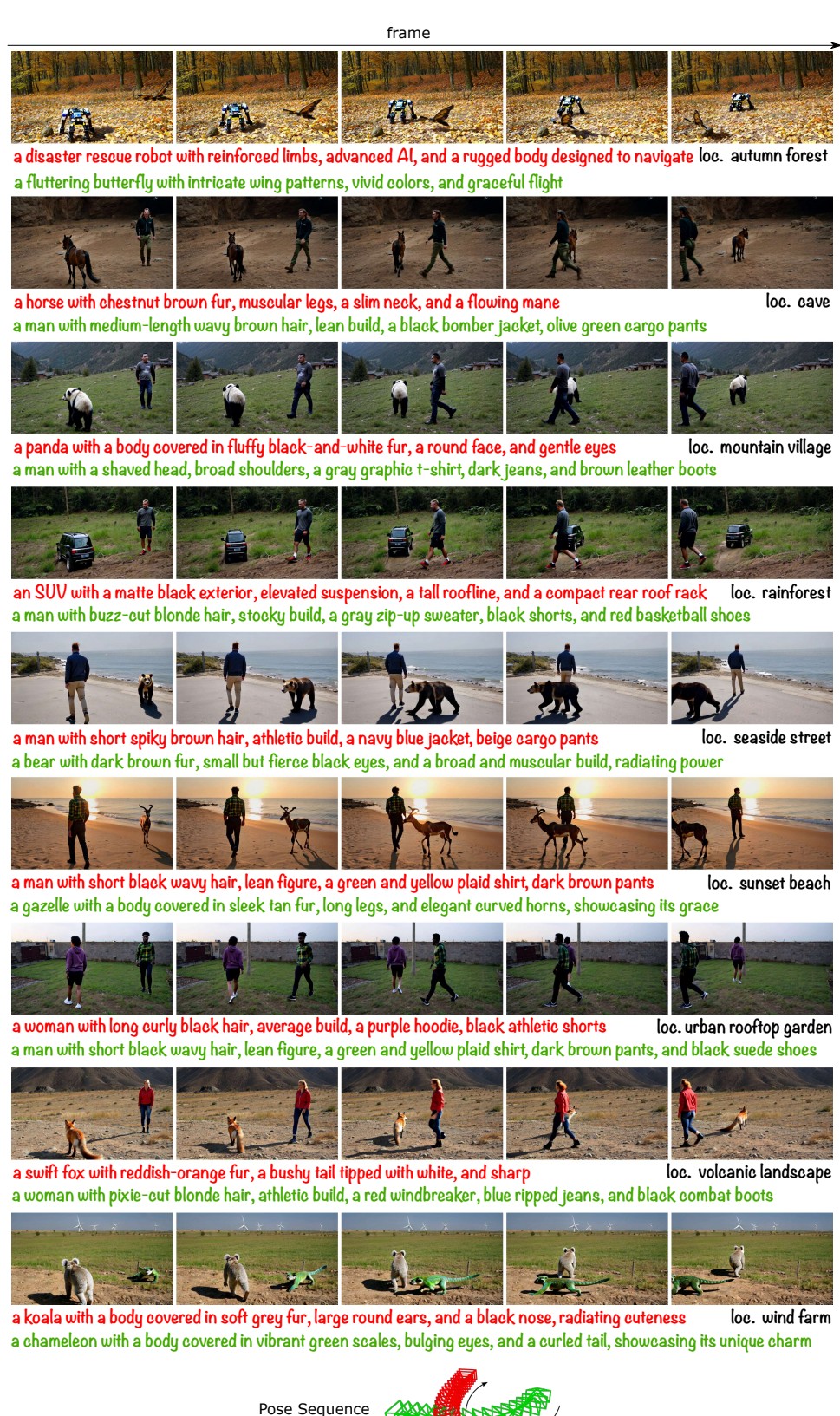

Figure S18: **Generalizable Results with Novel 3D Trajectories & Entity Prompts (7/20)**

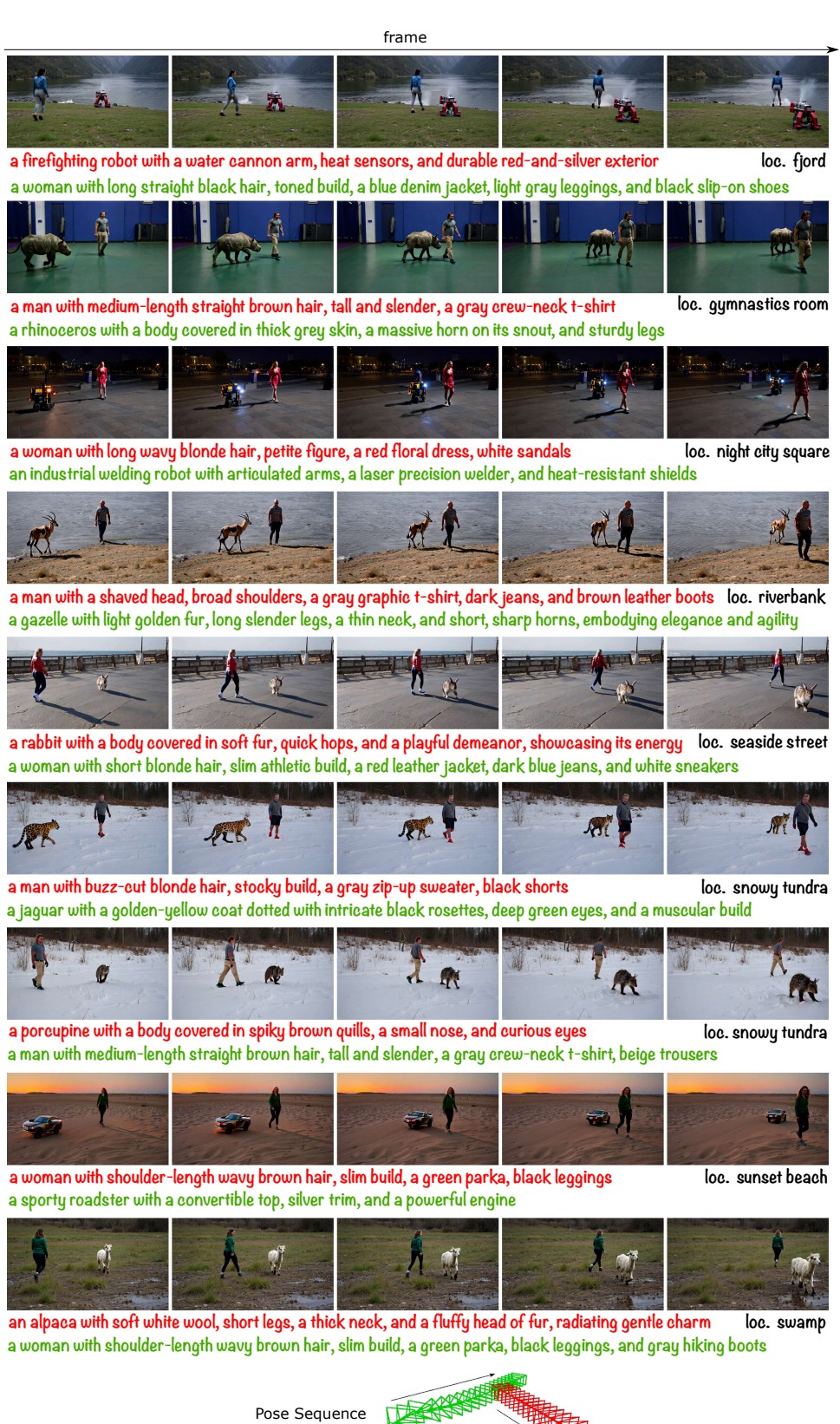

Figure S19: **Generalizable Results with Novel 3D Trajectories & Entity Prompts (8/20)**

frame

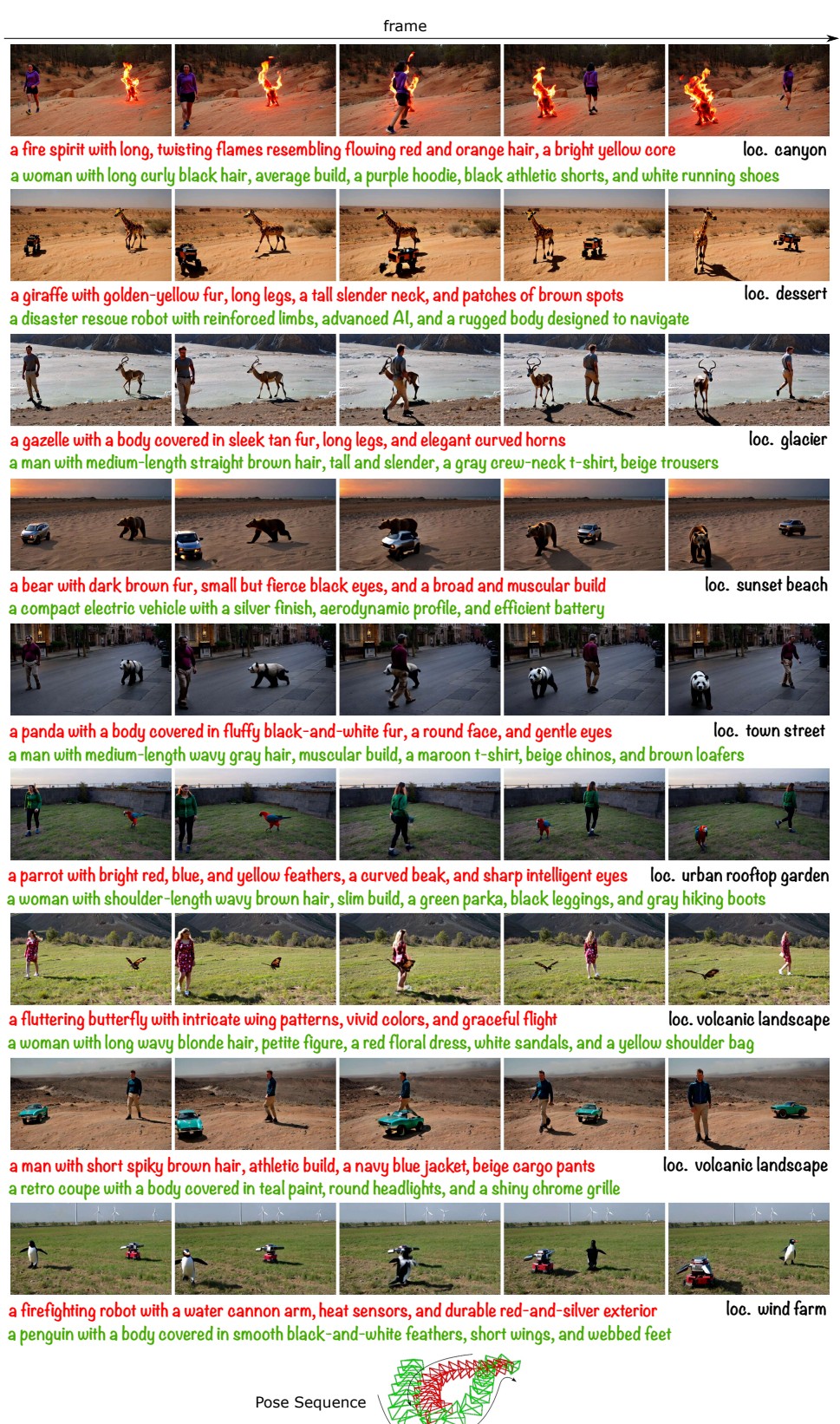

a fire spirit with long, twisting flames resembling flowing red and orange hair, a bright yellow core — loc. canyon
a woman with long curly black hair, average build, a purple hoodie, black athletic shorts, and white running shoes

a giraffe with golden-yellow fur, long legs, a tall slender neck, and patches of brown spots — loc. dessert
a disaster rescue robot with reinforced limbs, advanced AI, and a rugged body designed to navigate

a gazelle with a body covered in sleek tan fur, long legs, and elegant curved horns — loc. glacier
a man with medium-length straight brown hair, tall and slender, a gray crew-neck t-shirt, beige trousers

a bear with dark brown fur, small but fierce black eyes, and a broad and muscular build — loc. sunset beach
a compact electric vehicle with a silver finish, aerodynamic profile, and efficient battery

a panda with a body covered in fluffy black-and-white fur, a round face, and gentle eyes — loc. town street
a man with medium-length wavy gray hair, muscular build, a maroon t-shirt, beige chinos, and brown loafers

a parrot with bright red, blue, and yellow feathers, a curved beak, and sharp intelligent eyes — loc. urban rooftop garden
a woman with shoulder-length wavy brown hair, slim build, a green parka, black leggings, and gray hiking boots

a fluttering butterfly with intricate wing patterns, vivid colors, and graceful flight — loc. volcanic landscape
a woman with long wavy blonde hair, petite figure, a red floral dress, white sandals, and a yellow shoulder bag

a man with short spiky brown hair, athletic build, a navy blue jacket, beige cargo pants — loc. volcanic landscape
a retro coupe with a body covered in teal paint, round headlights, and a shiny chrome grille

a firefighting robot with a water cannon arm, heat sensors, and durable red-and-silver exterior — loc. wind farm
a penguin with a body covered in smooth black-and-white feathers, short wings, and webbed feet

Pose Sequence

Figure S20: **Generalizable Results with Novel 3D Trajectories & Entity Prompts (9/20)**

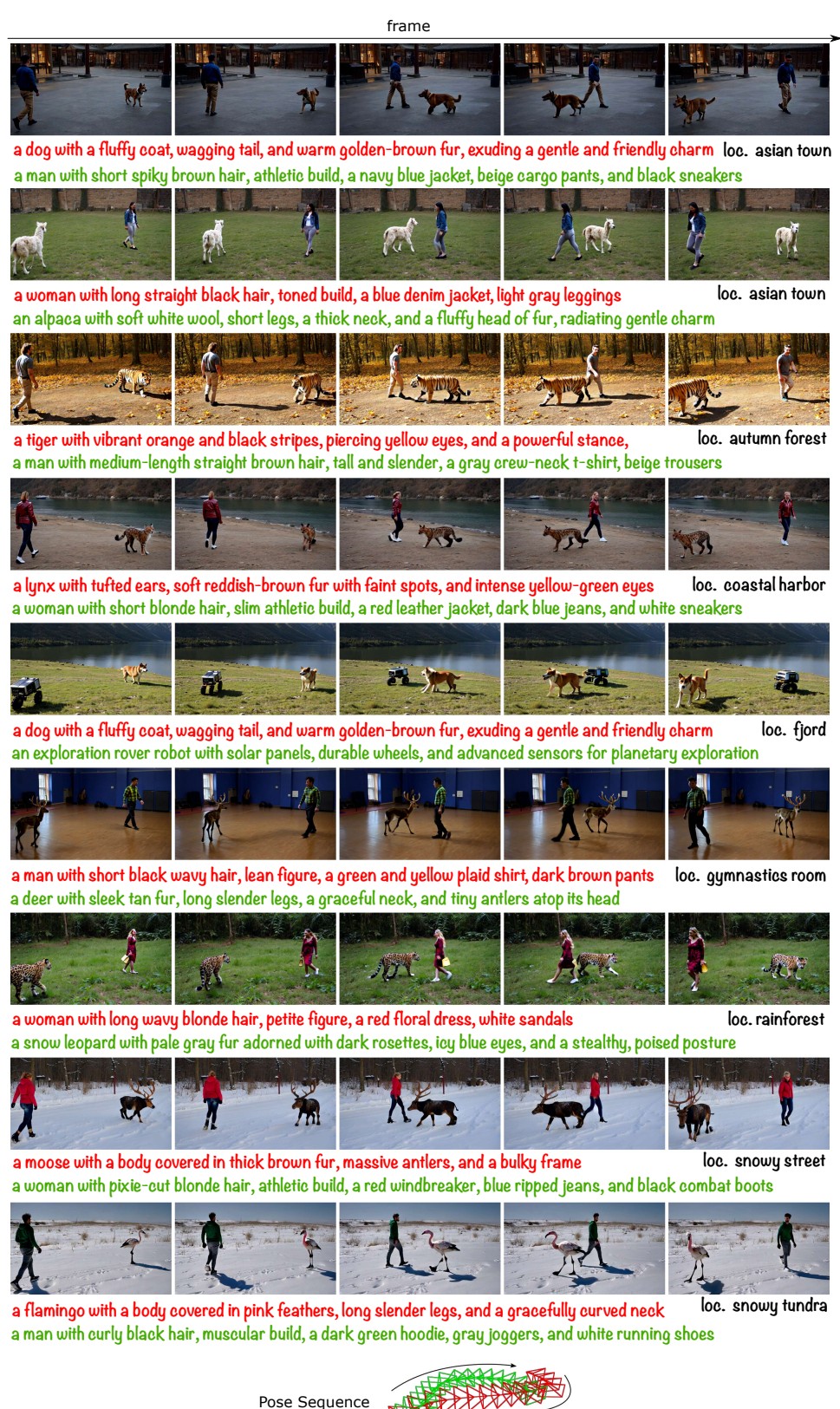

Figure S21: **Generalizable Results with Novel 3D Trajectories & Entity Prompts (10/20)**

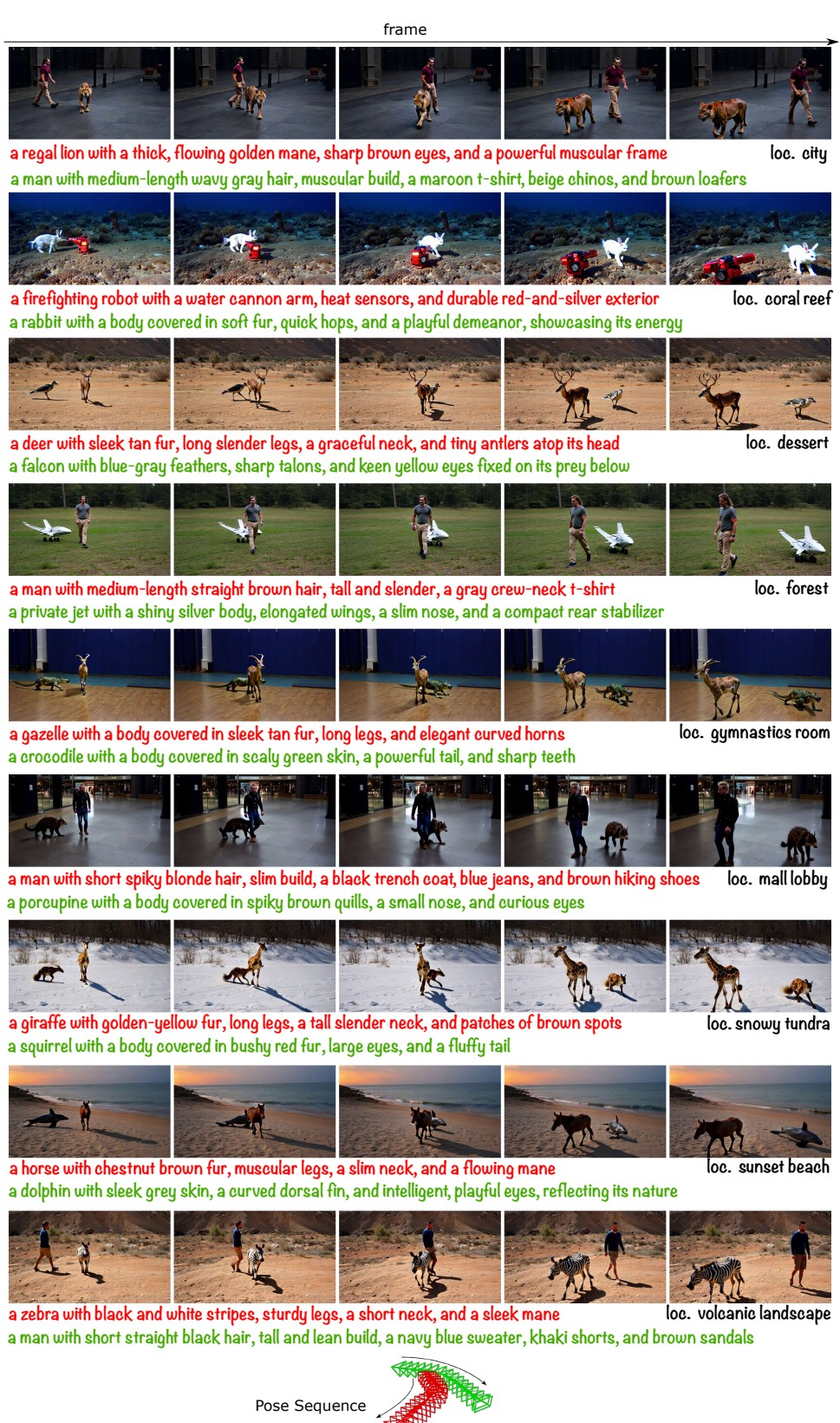

Figure S22: **Generalizable Results with Novel 3D Trajectories & Entity Prompts (11/20)**

frame

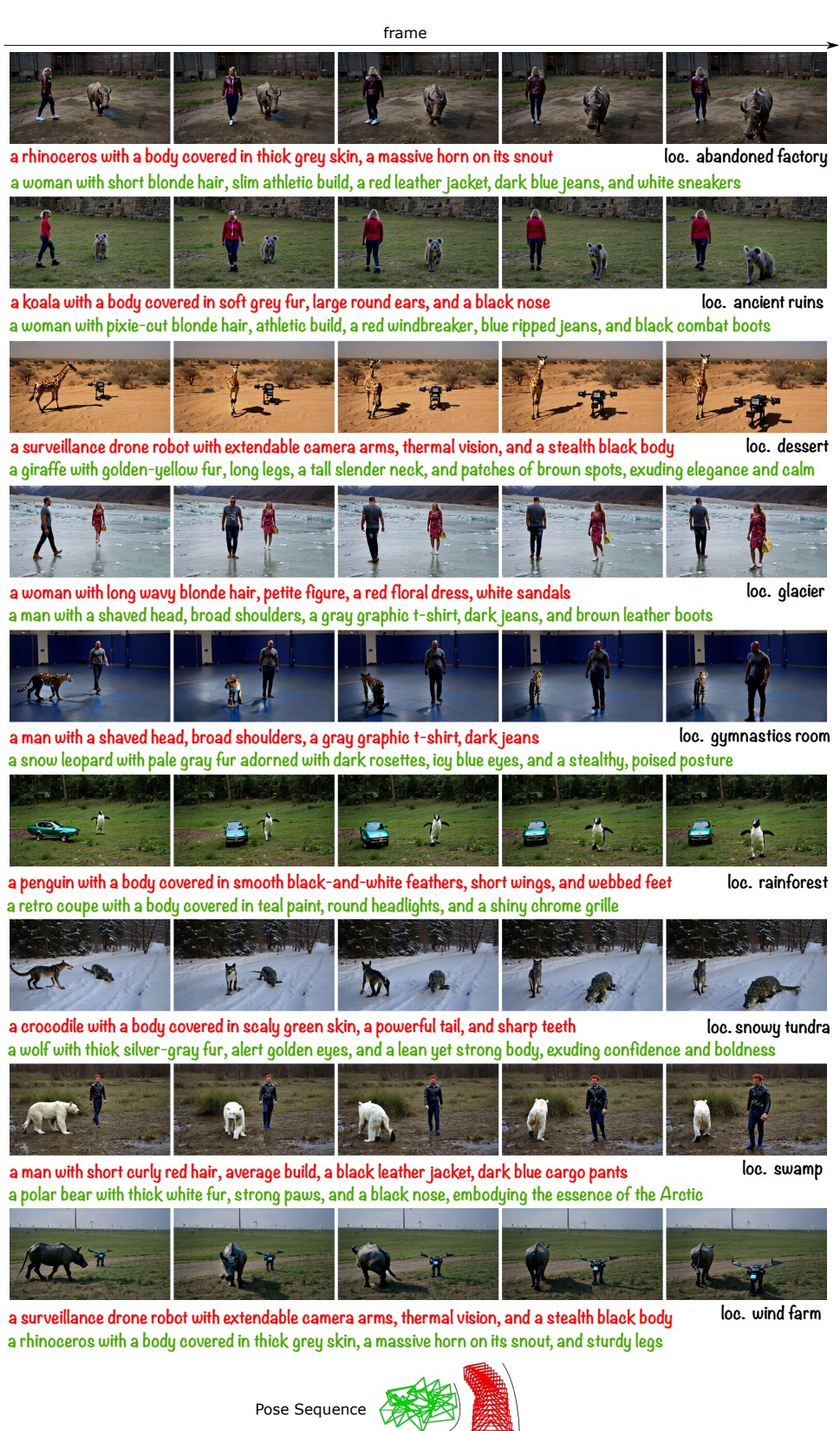

Figure S23: **Generalizable Results with Novel 3D Trajectories & Entity Prompts (12/20)**

frame

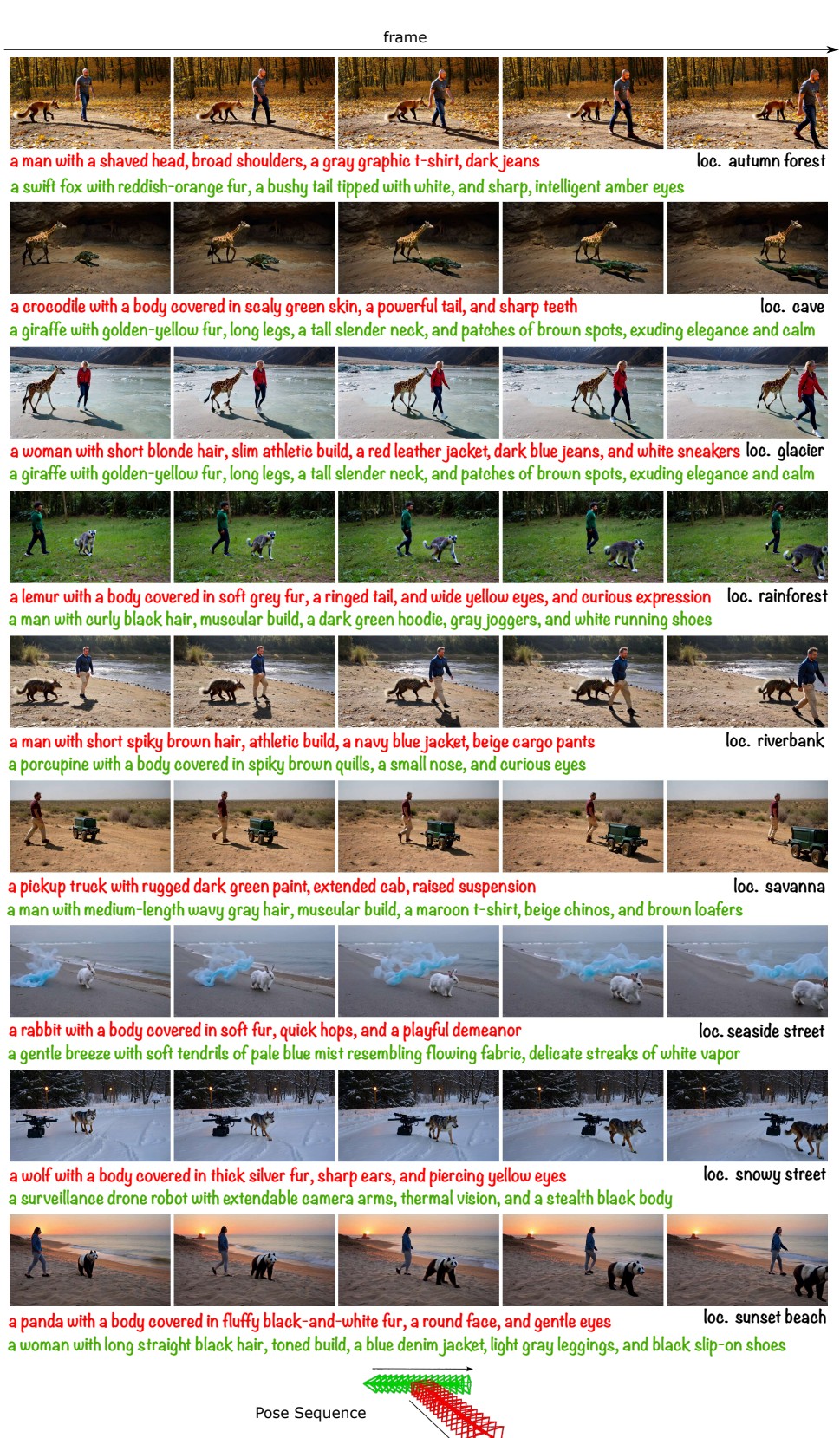

a man with a shaved head, broad shoulders, a gray graphic t-shirt, dark jeans — loc. autumn forest
a swift fox with reddish-orange fur, a bushy tail tipped with white, and sharp, intelligent amber eyes

a crocodile with a body covered in scaly green skin, a powerful tail, and sharp teeth — loc. cave
a giraffe with golden-yellow fur, long legs, a tall slender neck, and patches of brown spots, exuding elegance and calm

a woman with short blonde hair, slim athletic build, a red leather jacket, dark blue jeans, and white sneakers — loc. glacier
a giraffe with golden-yellow fur, long legs, a tall slender neck, and patches of brown spots, exuding elegance and calm

a lemur with a body covered in soft grey fur, a ringed tail, and wide yellow eyes, and curious expression — loc. rainforest
a man with curly black hair, muscular build, a dark green hoodie, gray joggers, and white running shoes

a man with short spiky brown hair, athletic build, a navy blue jacket, beige cargo pants — loc. riverbank
a porcupine with a body covered in spiky brown quills, a small nose, and curious eyes

a pickup truck with rugged dark green paint, extended cab, raised suspension — loc. savanna
a man with medium-length wavy gray hair, muscular build, a maroon t-shirt, beige chinos, and brown loafers

a rabbit with a body covered in soft fur, quick hops, and a playful demeanor — loc. seaside street
a gentle breeze with soft tendrils of pale blue mist resembling flowing fabric, delicate streaks of white vapor

a wolf with a body covered in thick silver fur, sharp ears, and piercing yellow eyes — loc. snowy street
a surveillance drone robot with extendable camera arms, thermal vision, and a stealth black body

a panda with a body covered in fluffy black-and-white fur, a round face, and gentle eyes — loc. sunset beach
a woman with long straight black hair, toned build, a blue denim jacket, light gray leggings, and black slip-on shoes

Pose Sequence

Figure S24: **Generalizable Results with Novel 3D Trajectories & Entity Prompts (13/20)**

frame

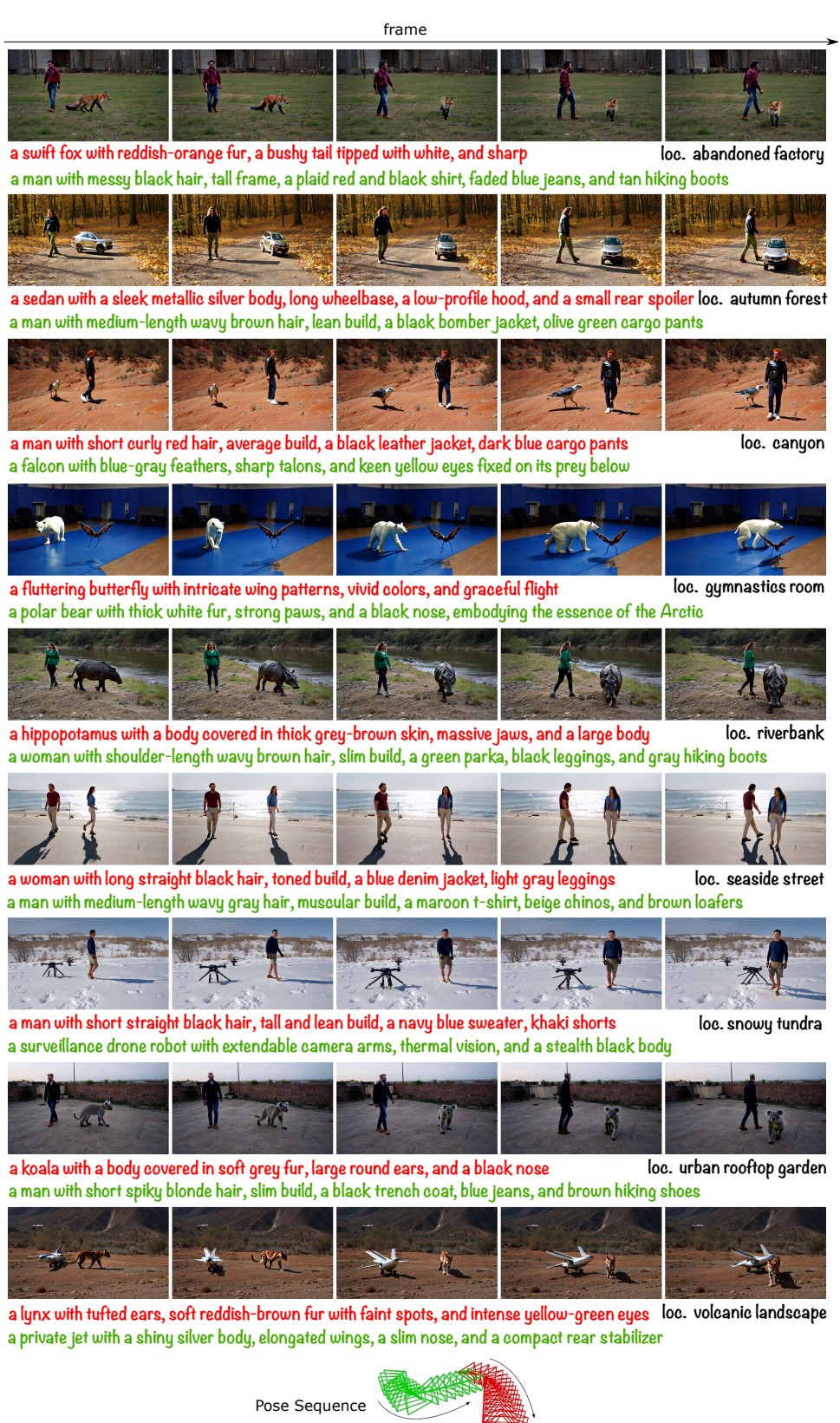

a swift fox with reddish-orange fur, a bushy tail tipped with white, and sharp | loc. abandoned factory
a man with messy black hair, tall frame, a plaid red and black shirt, faded blue jeans, and tan hiking boots

a sedan with a sleek metallic silver body, long wheelbase, a low-profile hood, and a small rear spoiler | loc. autumn forest
a man with medium-length wavy brown hair, lean build, a black bomber jacket, olive green cargo pants

a man with short curly red hair, average build, a black leather jacket, dark blue cargo pants | loc. canyon
a falcon with blue-gray feathers, sharp talons, and keen yellow eyes fixed on its prey below

a fluttering butterfly with intricate wing patterns, vivid colors, and graceful flight | loc. gymnastics room
a polar bear with thick white fur, strong paws, and a black nose, embodying the essence of the Arctic

a hippopotamus with a body covered in thick grey-brown skin, massive jaws, and a large body | loc. riverbank
a woman with shoulder-length wavy brown hair, slim build, a green parka, black leggings, and gray hiking boots

a woman with long straight black hair, toned build, a blue denim jacket, light gray leggings | loc. seaside street
a man with medium-length wavy gray hair, muscular build, a maroon t-shirt, beige chinos, and brown loafers

a man with short straight black hair, tall and lean build, a navy blue sweater, khaki shorts | loc. snowy tundra
a surveillance drone robot with extendable camera arms, thermal vision, and a stealth black body

a koala with a body covered in soft grey fur, large round ears, and a black nose | loc. urban rooftop garden
a man with short spiky blonde hair, slim build, a black trench coat, blue jeans, and brown hiking shoes

a lynx with tufted ears, soft reddish-brown fur with faint spots, and intense yellow-green eyes | loc. volcanic landscape
a private jet with a shiny silver body, elongated wings, a slim nose, and a compact rear stabilizer

Pose Sequence

Figure S25: **Generalizable Results with Novel 3D Trajectories & Entity Prompts (14/20)**

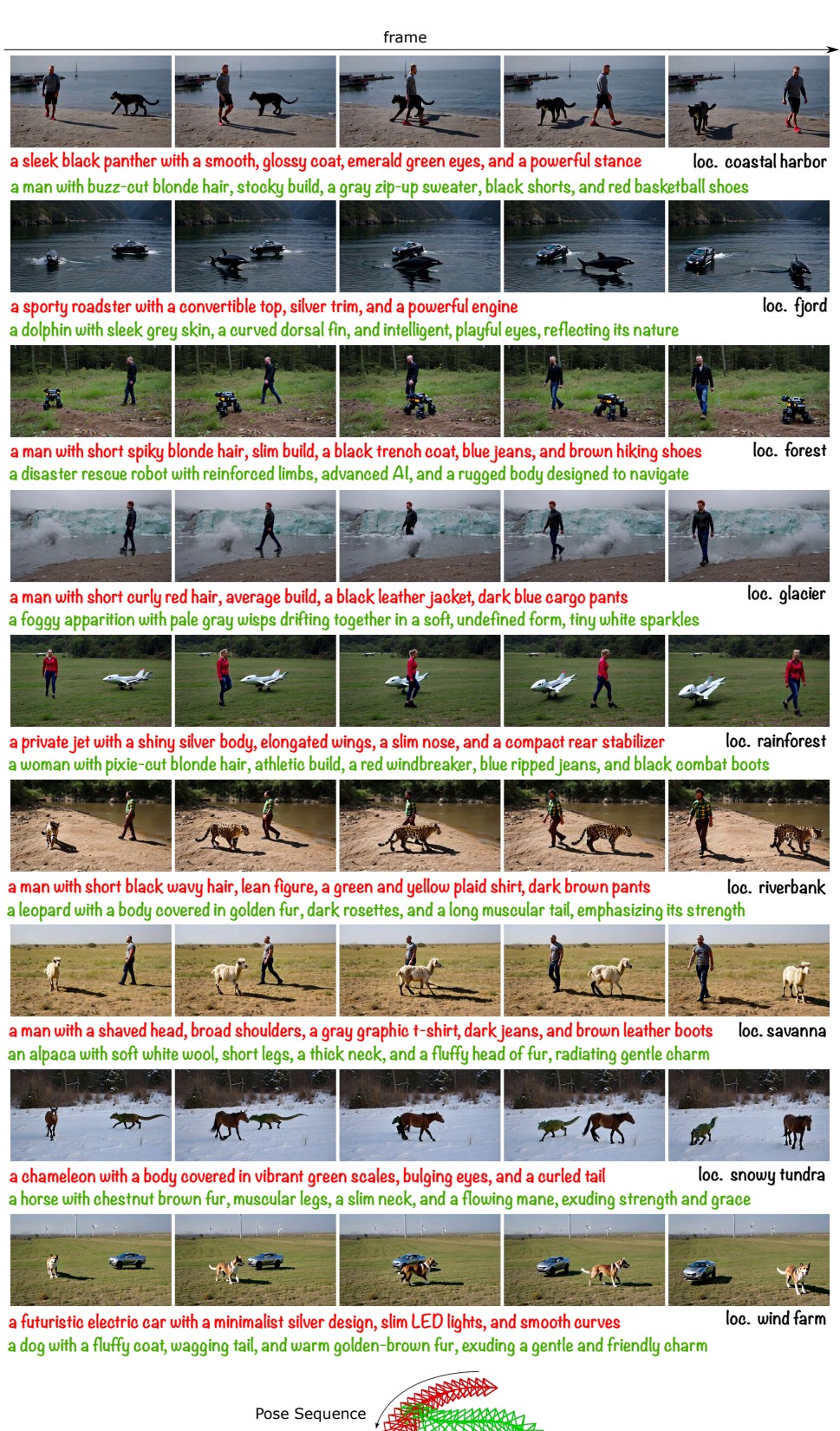

Figure S26: **Generalizable Results with Novel 3D Trajectories & Entity Prompts (15/20)**

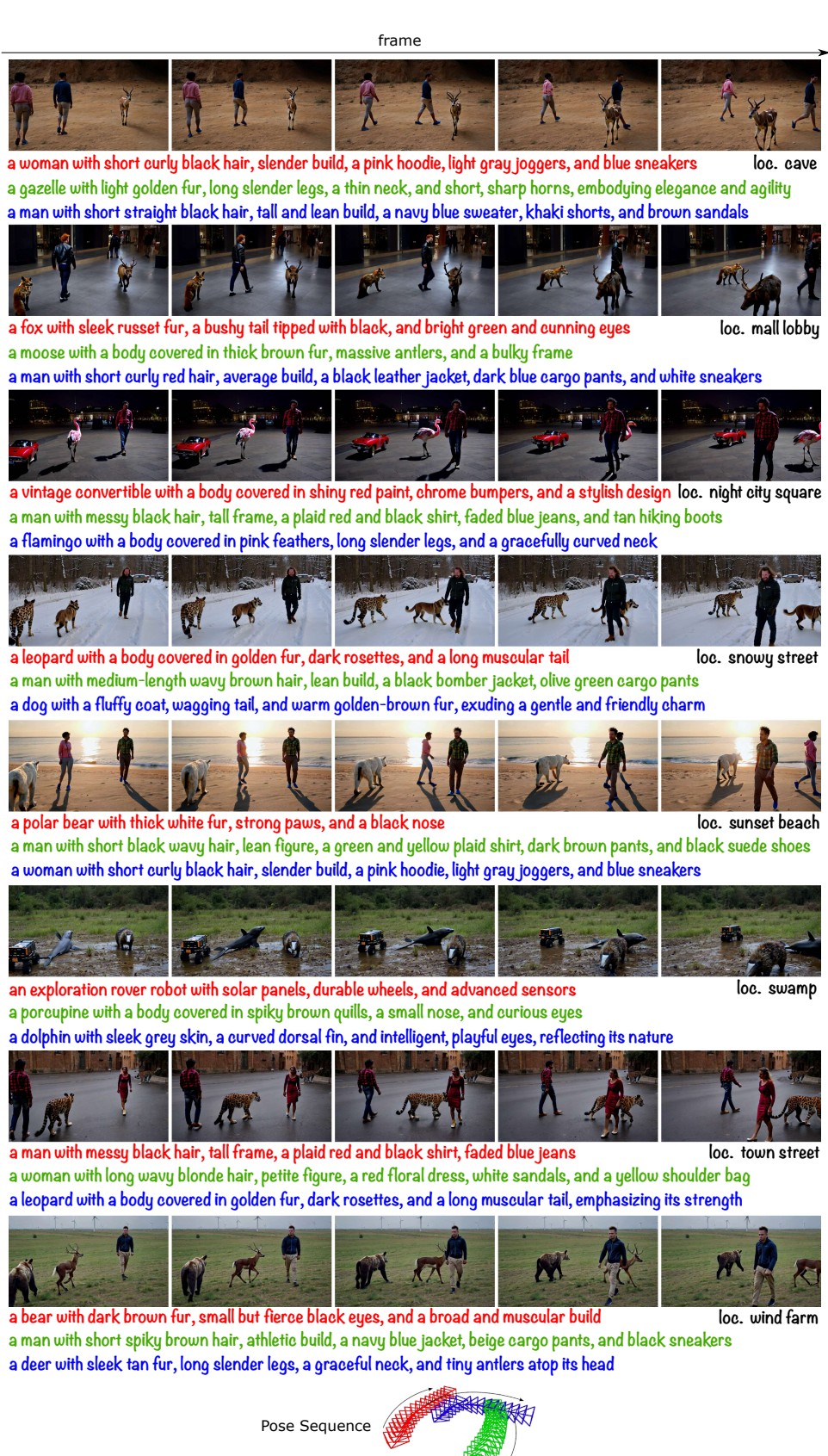

Figure S27: **Generalizable Results with Novel 3D Trajectories & Entity Prompts (16/20)**

frame

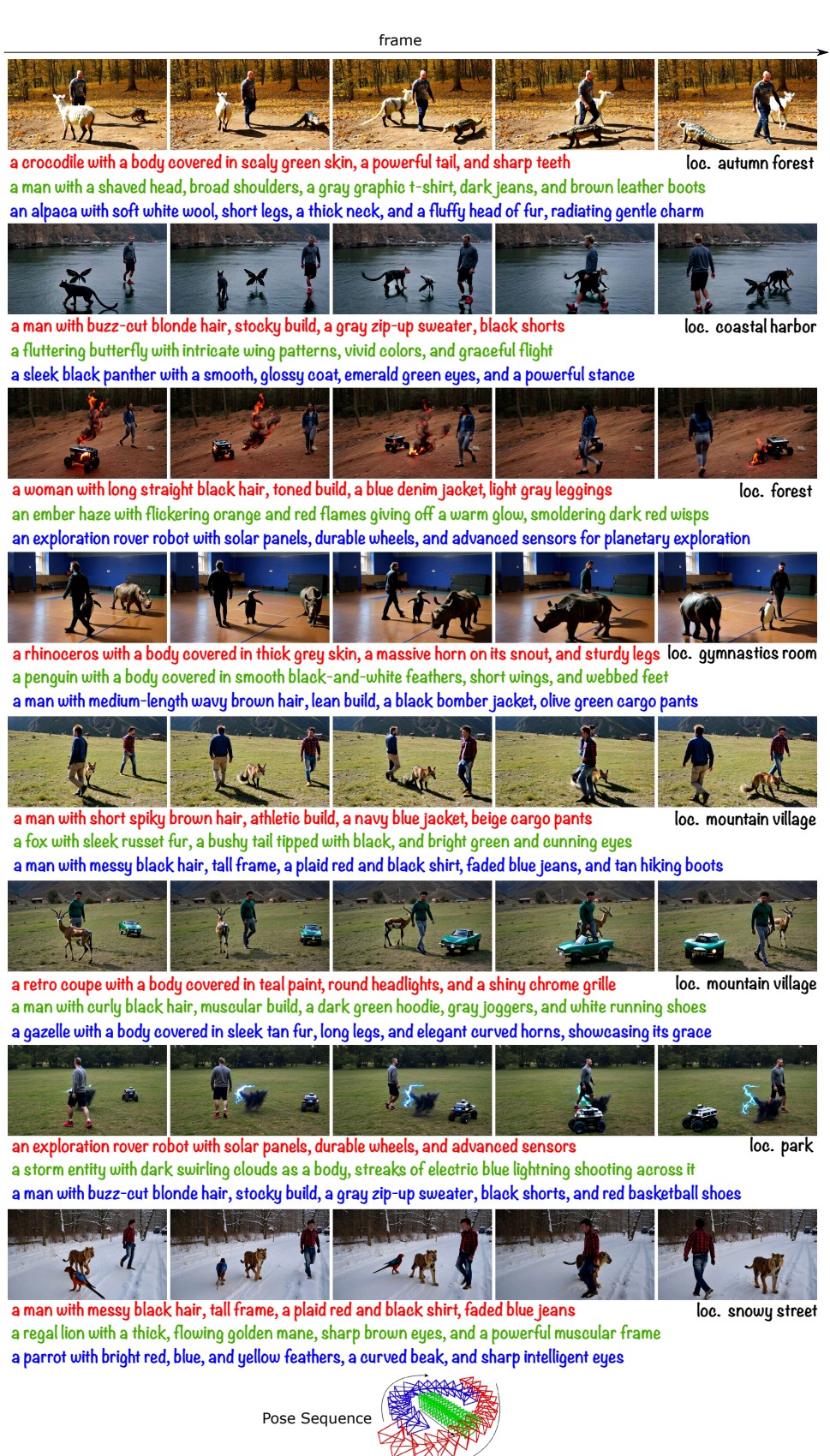

a crocodile with a body covered in scaly green skin, a powerful tail, and sharp teeth — loc. autumn forest
a man with a shaved head, broad shoulders, a gray graphic t-shirt, dark jeans, and brown leather boots
an alpaca with soft white wool, short legs, a thick neck, and a fluffy head of fur, radiating gentle charm

a man with buzz-cut blonde hair, stocky build, a gray zip-up sweater, black shorts — loc. coastal harbor
a fluttering butterfly with intricate wing patterns, vivid colors, and graceful flight
a sleek black panther with a smooth, glossy coat, emerald green eyes, and a powerful stance

a woman with long straight black hair, toned build, a blue denim jacket, light gray leggings — loc. forest
an ember haze with flickering orange and red flames giving off a warm glow, smoldering dark red wisps
an exploration rover robot with solar panels, durable wheels, and advanced sensors for planetary exploration

a rhinoceros with a body covered in thick grey skin, a massive horn on its snout, and sturdy legs — loc. gymnastics room
a penguin with a body covered in smooth black-and-white feathers, short wings, and webbed feet
a man with medium-length wavy brown hair, lean build, a black bomber jacket, olive green cargo pants

a man with short spiky brown hair, athletic build, a navy blue jacket, beige cargo pants — loc. mountain village
a fox with sleek russet fur, a bushy tail tipped with black, and bright green and cunning eyes
a man with messy black hair, tall frame, a plaid red and black shirt, faded blue jeans, and tan hiking boots

a retro coupe with a body covered in teal paint, round headlights, and a shiny chrome grille — loc. mountain village
a man with curly black hair, muscular build, a dark green hoodie, gray joggers, and white running shoes
a gazelle with a body covered in sleek tan fur, long legs, and elegant curved horns, showcasing its grace

an exploration rover robot with solar panels, durable wheels, and advanced sensors — loc. park
a storm entity with dark swirling clouds as a body, streaks of electric blue lightning shooting across it
a man with buzz-cut blonde hair, stocky build, a gray zip-up sweater, black shorts, and red basketball shoes

a man with messy black hair, tall frame, a plaid red and black shirt, faded blue jeans — loc. snowy street
a regal lion with a thick, flowing golden mane, sharp brown eyes, and a powerful muscular frame
a parrot with bright red, blue, and yellow feathers, a curved beak, and sharp intelligent eyes

Pose Sequence

Figure S28: **Generalizable Results with Novel 3D Trajectories & Entity Prompts (17/20)**

frame

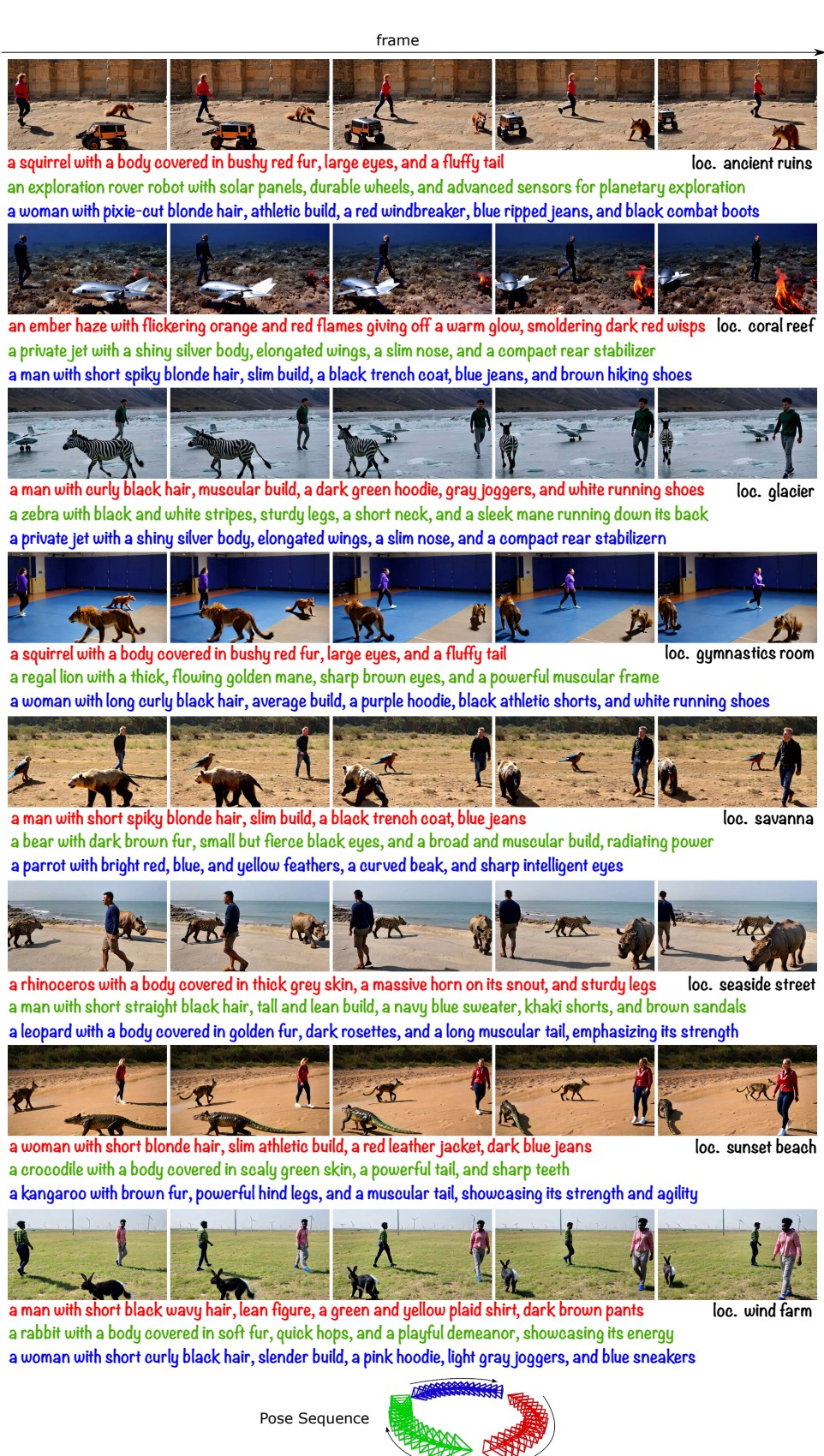

a squirrel with a body covered in bushy red fur, large eyes, and a fluffy tail
an exploration rover robot with solar panels, durable wheels, and advanced sensors for planetary exploration
a woman with pixie-cut blonde hair, athletic build, a red windbreaker, blue ripped jeans, and black combat boots
loc. ancient ruins

an ember haze with flickering orange and red flames giving off a warm glow, smoldering dark red wisps
a private jet with a shiny silver body, elongated wings, a slim nose, and a compact rear stabilizer
a man with short spiky blonde hair, slim build, a black trench coat, blue jeans, and brown hiking shoes
loc. coral reef

a man with curly black hair, muscular build, a dark green hoodie, gray joggers, and white running shoes
a zebra with black and white stripes, sturdy legs, a short neck, and a sleek mane running down its back
a private jet with a shiny silver body, elongated wings, a slim nose, and a compact rear stabilizern
loc. glacier

a squirrel with a body covered in bushy red fur, large eyes, and a fluffy tail
a regal lion with a thick, flowing golden mane, sharp brown eyes, and a powerful muscular frame
a woman with long curly black hair, average build, a purple hoodie, black athletic shorts, and white running shoes
loc. gymnastics room

a man with short spiky blonde hair, slim build, a black trench coat, blue jeans
a bear with dark brown fur, small but fierce black eyes, and a broad and muscular build, radiating power
a parrot with bright red, blue, and yellow feathers, a curved beak, and sharp intelligent eyes
loc. savanna

a rhinoceros with a body covered in thick grey skin, a massive horn on its snout, and sturdy legs
a man with short straight black hair, tall and lean build, a navy blue sweater, khaki shorts, and brown sandals
a leopard with a body covered in golden fur, dark rosettes, and a long muscular tail, emphasizing its strength
loc. seaside street

a woman with short blonde hair, slim athletic build, a red leather jacket, dark blue jeans
a crocodile with a body covered in scaly green skin, a powerful tail, and sharp teeth
a kangaroo with brown fur, powerful hind legs, and a muscular tail, showcasing its strength and agility
loc. sunset beach

a man with short black wavy hair, lean figure, a green and yellow plaid shirt, dark brown pants
a rabbit with a body covered in soft fur, quick hops, and a playful demeanor, showcasing its energy
a woman with short curly black hair, slender build, a pink hoodie, light gray joggers, and blue sneakers
loc. wind farm

Pose Sequence

Figure S29: **Generalizable Results with Novel 3D Trajectories & Entity Prompts (18/20)**

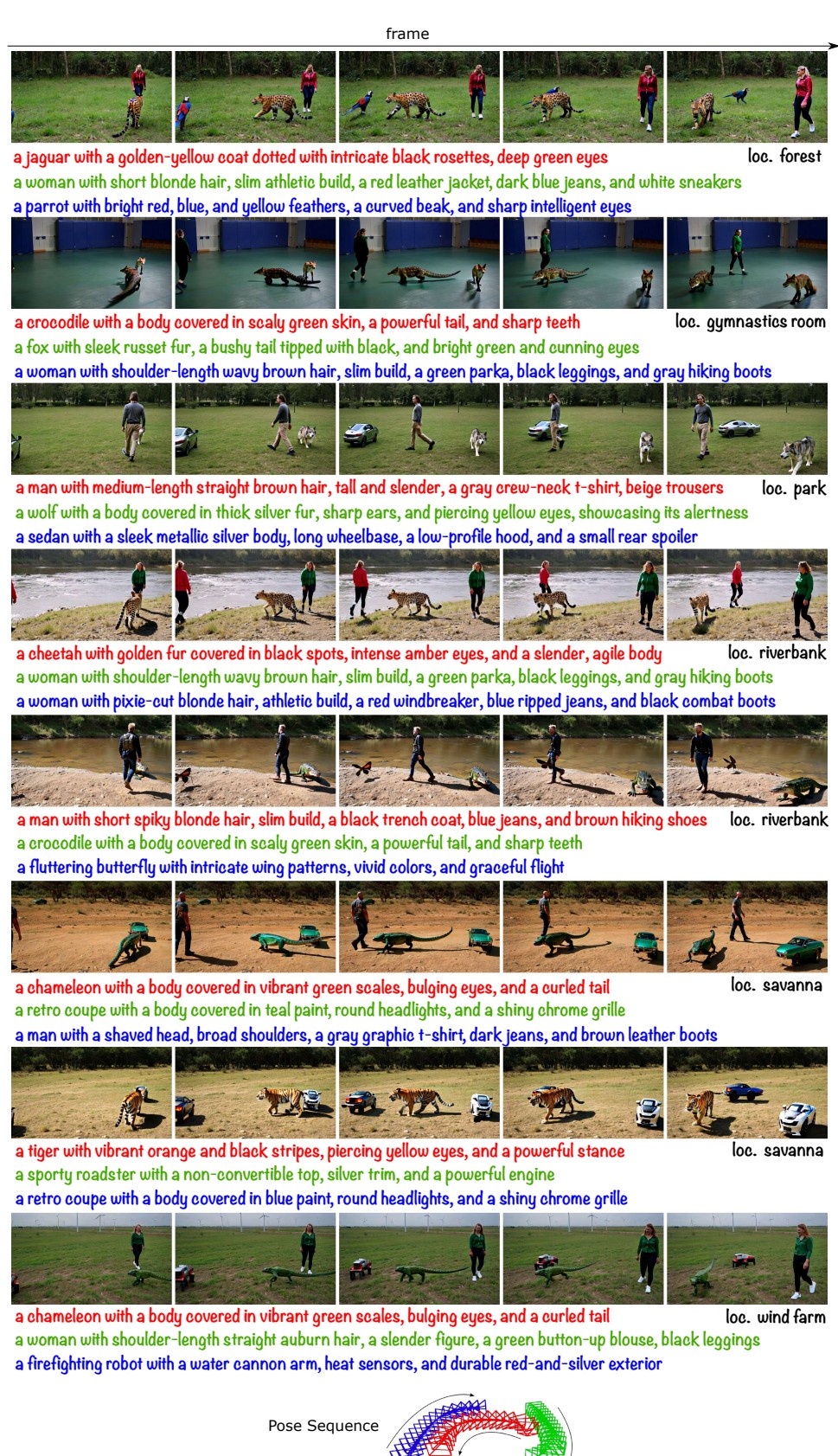

Figure S30: **Generalizable Results with Novel 3D Trajectories & Entity Prompts (19/20)**

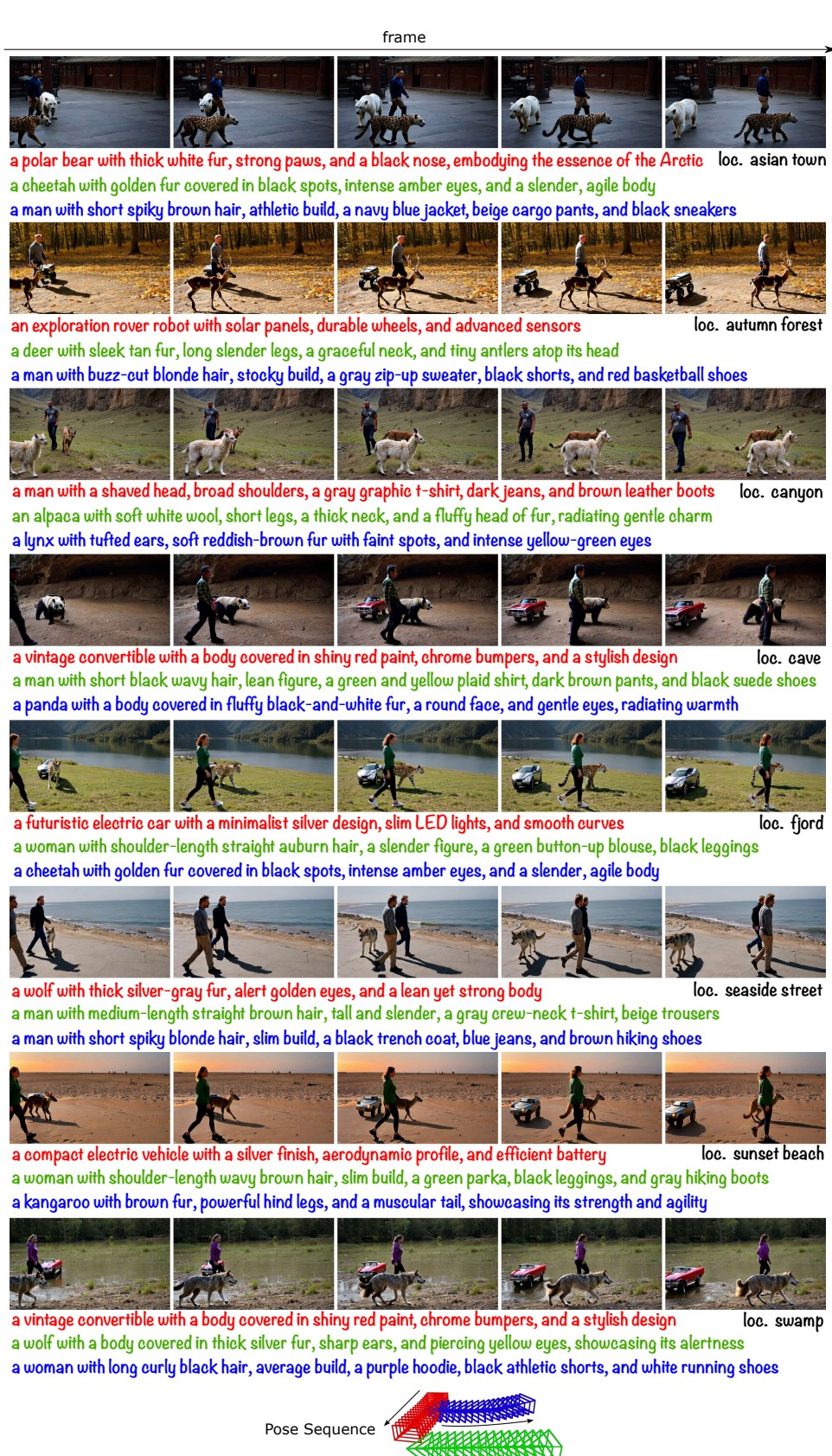

Figure S31: **Generalizable Results with Novel 3D Trajectories & Entity Prompts (20/20)**