# OpenReview forum: "3DTrajMaster: Mastering 3D Trajectory for Multi-Entity Motion in Video Generation"
_ICLR.cc/2025/Conference — ICLR 2025 Poster_

### Official Review · Reviewer_95Hu · 2024-10-29

**Soundness:** 3
**Presentation:** 3
**Contribution:** 3
**Rating:** 8
**Confidence:** 2

**Summary:**

Paper tackles the problem of multi-entity 3D control signals for video generation

Approach works by constructing an Unreal Engine dataset of assets with GPT4V generated trajectories and various camera angles, and then LoRAing it to do domain adaptation to prevent it from looking too much like the UE assets.

Extensive qualitative results indicate the method clearly works, and quantitative results indicate value of components + favorable results compared to several baselines.

**Strengths:**

The method clearly works. The supplemental provides extensive videos paired with prompts showcasing multiple walking agents. While generations are imperfect, I was shocked at the quality -- the assets seem to properly interact with light in the scenes, including full long shadows, as well as the terrain below, such as water.

The method used to achieve this adaptation seems straightfoward, and the recipe seems generally replicable and applicable to other video generation models despite the fact that the generation model used is proprietary.

**Weaknesses:**

- Pose evals are human only; however, I think this is well motivated given the stated lack of general video pose predictors.

Quite honestly, I am not an expert in this domain and I lack the background to provide meaningful criticism. The results look good, the actors clearly follow the given trajectories, and the recipe given to achieve this generally makes sense and feels general enough to be useful for arbitrary video generation models.

**Questions:**

- Why does the alligator gait look so much worse than the other animals? Its legs seem to be almost static in some scenes.
 - It seems that the proprietary base video generation model is extremely strong (congrats!). Do you think this recipe will work with less strong video generation models, such as current open source models?

---

> ### Author Response · Authors · 2024-11-16
> **Response to Reviewer 95Hu**
>
> Thank you for your genuine feedback! We kindly urge you to check our "Update on Warm-up Discussion, Anonymous Website, Paper Revision, and Dataset&Code Release" to catch up on our latest updates.
>
> Moreover, we address your concerns below.
>
> **(Q1 Poor Examples of “alligator gait”)**
>
> We provide additional examples of alligators with normal gait on our anonymous website (please refer to "Response to Reviewer 95Hu" below the website for more details). Regarding the poor quality of the supplementary files, we attribute this to two factors: 1) randomness in video generation, as all our supplementary videos are generated only once with a single seed, and 2) the relatively low-quality animation of the quadrupedal dinosaurs (you can check some samples in "360Motion Dataset Subset (480 videos)" on the website) during the training phase, which have large body sizes and resemble alligator to some extent.
>
> If you would like to generate alligator videos with specific trajectories, or if you have any concerns regarding other entities, please don't hesitate to let us know.
>
> **(Q2 Applied on Less Strong Video Generation Models)**
>
> We are currently working on adapting our design to the publicly available CogVideoX-5B model. To showcase our progress, we have rendered videos in 480x720 resolution, which is compatible with CogVideoX-5B (please see some samples in "Response for Reviewer 95Hu" on the website). We believe that our plug-and-play module can also be applied to other video foundation models, including those with less prior.
>
> If you have further questions or expected experiments, feel free to let us know. Besides, all reviewers are welcome to request extra video samples with specific entity prompts, trajectory templates, and locations, and we will generate those videos and make them available on the website.

---

> ### Author Response · Authors · 2024-11-24
> **Awaiting Your Response (3 Days Left)**
>
> Dear Reviewer `95Hu`,
>
> Thank you once again for your time and effort in reviewing our work. With only `3 days` remaining, we kindly request your feedback on our response. If any part of our explanation is unclear or if you would like further details on any aspect, please don't hesitate to reach out.
>
> We are eager to ensure everything is addressed to your satisfaction and are happy to provide additional clarification promptly before the discussion deadline.
>
> Best,
>
> 3DTrajMaster Author(s) Team

---

> ### Author Response · Authors · 2024-12-03
> **Thanks for Your Recognization!**
>
> Dear Reviewer `95Hu`,
>
> Thanks again for your efforts in reviewing this work and providing such a positive review. We greatly appreciate your valuable feedback, which include:
>
> 1. Compliments on our impressive results and the quality of generalizable video samples. Our goal is to learn high-quality 3D motion patterns and extend them to real-world quality using the power of video foundation models.
>
> 2. Your concerns regarding the specific animal sample have prompted us to further evaluate the dataset’s quality and address potential issues with animal animations during training. Your feedback also highlights corner cases to consider when generalizing to a broader range of samples.
>
> 3. Your inquiry about applying our approach to other open-source video models, such as CogVideoX and Mochi 1 preview, motivates us to extend our plug-and-play models to additional frameworks (currently ongoing), broadening our impact within the research community.
>
> We really appreciate it and we believe this work will motivate more follow-up research to `explore fine-grained 3D-aware motion control in video generation`.
>
> Best,
>
> 3DTrajMaster Author(s) Team

---

### Official Review · Reviewer_2Hag · 2024-10-31

**Soundness:** 3
**Presentation:** 3
**Contribution:** 3
**Rating:** 8
**Confidence:** 4

**Summary:**

1. This paper aims to control the entities’ motion with 3D control signals in video generation.  3D control signals are a more natural representation compared to 2D signals as the motion is in 3D space.
2. This paper proposed a plug-and-play module to integrate the entities with their respective 3D trajectories into a pre-trained video generative model to control the 3D motion.
3. To avoid video quality degradation during the fine-tuning, they use a Lora-like domain adaptor and an annealed sampling strategy.
4. They construct a synthetic dataset collecting dynamic 3D human and animal assets with ground-truth 3D motion for training.
5. Experiments show that the proposed methods can achieve state-of-the-art performance in 3D motion control.

**Strengths:**

1. The proposed method is the first to control entities’ motion with 3D trajectories in video generation. The task is novel and reasonable as 3D control signals can fully express the inherent 3D nature of motion and offer better controllability in video generation compared to 2D control signals.
2. The method design is clear and reasonable.
3. The paper constructs a new synthetic dataset for this task. The dataset potentially benefits the following video generation with 3D entity control.
4. The experiments are thorough and solid. Plenty of visualization results as well as the videos in the supplementary demonstrate the effectiveness and generability of the proposed method.
5. The paper is well-written and easy to follow.

**Weaknesses:**

1. The dataset lacks diversity in terms of background and motion types. The setting is restricted to a "City" environment (as noted in the paper's Limitations section), and the actions are primarily limited to walking. Consequently, models trained on this dataset are also constrained in their generalizability.
2. Foot skating/floating issues are prevalent in the dataset. This appears to result from inconsistencies between the relative motion and global motion of the dynamic entities, which could negatively impact model training by introducing artifacts.

Minor:

1. The explanation of the "ControlNet-like architecture" is vague and lacks clarity. The paper references this term in Lines 20 and 115, suggesting it pertains to the Object Injector, whose initial weights stem from the 2D spatial self-attention layer in the video generative model. However, this does not align with a true ControlNet-like module, which would typically function as a parallel module with zero-initialized layers designed to adjust the original features. Instead, the Object Injector is positioned after the 2D spatial self-attention layer, differing fundamentally from ControlNet-like behavior.
2. It would have been better to elaborate on Line 334: “This phenomenon reflects the model’s ability to learn 3D motion representations”? The reasoning behind this statement isn’t entirely clear.

**Questions:**

In Table 2, could you clarify the distinction between “Multiple Entities” and “All Entities”? Does “All Entities” include both “single” and “multiple” cases?

---

> ### Author Response · Authors · 2024-11-16
> **Response to Reviewer 2Hag**
>
> Thanks for your insightful feedback! We address your concerns below. If you have further questions or expected experiments, feel free to let us know. Besides, all reviewers are welcome to request extra video samples with specific entity prompts, trajectory templates, and locations, and we will generate those videos and make them available on the website.
>
> **(W1 Lacking Diversity in Dataset)**
>
> **1. Background**
>
>  We have extended the pure city dataset to more diverse 3D UE scenes such as desert, forest, Asian town, HDRIs (projected into 3D), as well as more complex 3D trajectory templates. Please see the dataset samples on our [anonymous website](https://3dtrajmaster.github.io/). Below is a summary of the differences.
>
> | Dataset      | 3D Scenes | Samples     |  3D Trajectory Templates     |
> | --- | --- | :---: |:---: |
> | Rebuttal Version   | 9 (city, desert, forest, Asian town, 5 HDRIs)       | 54,000     | 96  |
> | Submission Version     | 1 (city)       | 12,000   | 48  |
>
> **2. Motion Type**
>
> We acknowledge that our current motion types are limited to walking or simple jumping, and we have incorporated this limitation in the "Limitation" section of the paper. However, motion encompasses a broad spectrum of types, including local motion (e.g., human dancing, waving hands, or leaves swaying in the wind), interactions (e.g., a person picking up a dog or a ball), as well as global motion (e.g., walking, running). Modeling all these motion patterns in a single work is challenging. Previous works have also focused on specific types of entity motion, such as Direct-a-video [1] and TC4D [2] for global motion, Generative Image Dynamics [3] for interactive local motion, and numerous works dedicated to human local motion. MotionCtrl and Tora attempt to model unified entity motion (global, local, and interactive) using 2D trajectory inputs. However, these methods lack entity-specific motion correspondence, which can lead to ambiguity when targeting particular entities or motion types (as discussed in our paper). We believe, however, that all of these motion patterns can be modeled similarly to our 6 DoF motions with structured motion patterns, where the key factor lies in the quality of the structured data.
>
> **(W2 Foot Skating/Floating Issues)**
>
> We admit these issues in constructing our dataset since it is hard to generate real-world-like video samples in pure UE engines. However, our key insight is to learn high-quality 3D motion patterns in synthetic data and alleviate these "foot floating" issues when generalizing to real-world scenarios through the power of video foundation models.
>
> **(W3-Minor Explanation of the "ControlNet-like architecture")**
>
> Thank you for pointing out this typo! We have corrected it to "injector architecture" in Line 20 and "plug-and-play architecture" in Line 115 in the latest version. Initially, we did try a ControlNet-like architecture, but the current design proved to be more effective. We wrote the paper early and inadvertently overlooked updating the description.
>
> **(W4-Minor Elaboration on Line 334)**
>
> We have modified the original context in our latest version. Here is the explanation: Since we do not randomly drop out motion sequences during training like text, the model implicitly learns static motion modeling from videos where entities are primarily in motion. Thus when setting static motion as a "negative motion prompt”, we can amplify the magnitude of movement, leading to improved pose accuracy during testing. This effect was observed when we trained CameraCtrl [4] on the RealEstate10K dataset with our internal DiT-based model. RealEstate10K consists entirely of camera-dynamic scenes, and during inference, we discovered that setting the negative camera embedding to static motions significantly enhances camera movement. Interestingly, this approach also improves the motion of 3D entities.
>
> **(Q1 Clarification of the Distinction between “Multiple Entities” and “All Entities”)**
>
> “Multiple Entities” contains 2 or 3 entities (88 samples), and “All Entities” contains all “single” and “multiple” cases (full 100 samples). Please refer to our benchmark results on the anonymous website.
>
> | Num. of All Entities | Num. of 1 Entity  | Num. of 2 Entities |  Num. of 3 Entities |
> | :---: | :---: | :---: |:---: |
> |  100 | 12  | 72 |  16 |
>
> [1] Direct-a-Video: Customized Video Generation with User-Directed Camera Movement and Object Motion. SIGGRAPH 2024
>
> [2] TC4D: Trajectory-Conditioned Text-to-4D Generation. ECCV 2024
>
> [3] Generative Image Dynamics. CVPR 2024
>
> [4] CameraCtrl: Enabling Camera Control for Text-to-Video Generation. ARXIV 2024

---

> > ### Comment · Reviewer_2Hag · 2024-11-23
> >
> > I appreciate the author's response. They have extended the dataset to include more diverse scenes and trajectories and have revised the previously inappropriate sentences. The response has addressed my concerns, so I will maintain my previous score.

---

> ### Author Response · Authors · 2024-11-23
> **Thanks for Your Recognization!**
>
> Dear Reviewer `2Hag`,
>
> Thanks again for your efforts in reviewing this work and providing such a positive review. We greatly appreciate your valuable suggestions, which include:
>
> 1. The question about the scaling of the dataset (specifically the background) has prompted us to better demonstrate the scalability of our proposed dataset construction approach and the potential generalizability of our model.
>
> 2. The observation regarding the limited scope of motion types and issues related to foot skating/floating has helped us clarify our design motivation and the current limitations more effectively.
>
> 3. The inquiry about improving the explanation of the 'ControlNet-like architecture' and the 'model’s ability to learn 3D motion representations' has helped us refine the quality of the paper.
>
>
> We really appreciate it and we believe this work will motivate more follow-up research to `explore fine-grained 3D-aware motion control in video generation`.
>
> Best,
>
> 3DTrajMaster author(s) team

---

### Official Review · Reviewer_WQpa · 2024-11-03

**Soundness:** 3
**Presentation:** 3
**Contribution:** 3
**Rating:** 5
**Confidence:** 3

**Summary:**

3DTrajMaster" introduces a method for controlling multi-entity 3D motion in video generation using 6DoF pose sequences. The authors propose a novel plug-and-play 3D-motion grounded object injector that fuses entity descriptions with corresponding 3D trajectories using a gated self-attention mechanism. They address the lack of suitable training data by constructing a 360°-Motion Dataset, combining collected 3D assets with GPT-generated trajectories. The method is tested against prior 2D motion control approaches and shows state-of-the-art performance in both motion control accuracy and generalization.

**Strengths:**

The proposed 3D-motion grounded object injector, combining 6DoF pose sequences with entity descriptions, is an innovative contribution that extends beyond 2D control limitations.

**Dataset Creation**: The construction of the 360°-Motion Dataset addresses a notable gap in available training data, particularly for multi-entity scenarios, using an innovative combination of GPT and UE.

**Flexibility**: The plug-and-play nature of the proposed object injector facilitates broader applicability across different generative models, with the gated self-attention mechanism ensuring entity-specific trajectory adherence.

**Weaknesses:**

**Dataset Limitation**: The reliance on synthetic data and a limited number of assets may hinder real-world generalization. The "city" setting constraint for the dataset also limits the diversity of possible outputs.

**Generalizability**: The model's performance for generalized 3D scenes beyond those captured in the MatrixCity platform remains unclear. More evaluation of real-world, diverse datasets would strengthen the contributions.

**Evaluation Scope**: While evaluation metrics like FVD and CLIP Similarity are used, the lack of real-world evaluations or qualitative feedback from human users makes it hard to gauge practical effectiveness fully. Also, the author could consider comparing with recent 4D generation methods such as TC4D which also can control trajectory.

I will also check other reviewers's feedback.

**Questions:**

Will the dataset and code will be public after acceptance?

---

> ### Author Response · Authors · 2024-11-15
> **Response to Reviewer WQpa (Part 1/3)**
>
> Thanks for your thoughtful feedback and the time you've dedicated to reviewing our work! As the proposed question will be a concern for all the reviewers, I have included it in the official review, please see "Update on Warm-up Discussion, Anonymous Website, Paper Revision, and Dataset&Code Release".
>
> Moreover, we address your concerns below.
>
> **(Q1 Dataset Release)**: Yes. We plan to release our dataset to support the research community once the paper is accepted, or possibly even earlier if the review scores are favorable. To demonstrate our commitment, we have provided a subset of smaller datasets on our anonymous website (including 5 scenes and 480 videos).
>
> **(Q2 Code Release)**: Yes, the release of the code based on our internal model will go through our internal license review process. However, we commit to releasing a version based on the publicly available CogVideoX once the paper is accepted, or potentially even earlier if the review scores are favorable. Additionally, all reviewers are welcome to request extra video samples with specific entity prompts, trajectory templates, and locations, and we will generate those videos and make them available on the website.

---

> ### Author Response · Authors · 2024-11-16
> **Response to Reviewer WQpa (Part 2/3)**
>
> **(W1 Dataset Limitation):**
>
> **1. Limited Data:** We have extended the pure city dataset to more diverse 3D UE scenes such as desert, forest, Asian town, HDRIs (projected into 3D), as well as more complex 3D trajectory templates. Please see the dataset samples on our [anonymous website](https://3dtrajmaster.github.io/). Below is a summary of the differences.
>
> | Dataset      | 3D Scenes | Samples     |  3D Trajectory Templates     |
> | --- | --- | :---: |:---: |
> | Rebuttal Version   | 9 (city, desert, forest, Asian town, 5 HDRIs)       | 54,000     | 96  |
> | Submission Version     | 1 (city)       | 12,000   | 48  |
>
> **2. Limited Number of Assets:** First, we fully agree that scaling up to include more entities could further enhance the model generalizability. However, we think that the current set of 70 entities is already sufficient to generate a wide variety of diverse entities. Please refer to 'Control Entity Motion with Diverse Entities' and 'Control Multi-Entity Motion with Complex 3D Trajectories' on our website, where we can even generalize to control the motion of natural forces, such as fire and clouds, in 3D space.
>
> **3.  Reliance on Synthetic Data:** We respectfully argue that this is the main contribution of our paper: training solely on synthetic data to learn high-quality 3D motion patterns and generalizing to real-world scenarios through the power of video foundation models. We have discussed the challenges of capturing high-quality real-world 6 DoF training data in Sec. 3.3 and outlined our motivation behind the data construction approach. To better adapt to real-world conditions, we propose the video domain adaptor and an annealed sampling strategy as additional technical contributions.  Please refer to our website for samples of real-world quality videos.
>
> **(W2 Generalizability):**
>
> **1. Benchmark Evaluation:**
> Compared to the previous evaluation with a single city location, we now support 32 diverse locations (please refer to our benchmark results on the website). Below is a summary of the differences, and for more details, please see Tables R4-R7.
>
> | Eval. Benchmark | Num. of Human Prompts  | Num. of Non-Human Prompts |  Num. of Location Prompts  |
> | --- | :---: | :---: |:---: |
> |  Rebuttal Version   | 20    | 52 (animal, car, robot)    |  32 |
> | Submission Version      | 20     | 40 (animal)  |  1 (city) |
>
> **2. Generalization Scenario:**
>
> Please see "Control Multi-Entity Motion with Complex 3D trajectories" (180 videos) on our anonymous website or Fig. S12-S31 in the paper.

---

> ### Author Response · Authors · 2024-11-19
> **Response to Reviewer WQpa (Part 3/3)**
>
> **(W3 Evaluation Scope)**
>
> **1. Comparison with TC4D**
>
> Thank you for highlighting this related work! While both TC4D and our approach require ground-truth 3D trajectories for generation, TC4D is still not an appropriate baseline as it focuses on compositional 4D scene generation, whereas our method is designed for video generation. We show the comparison in the following table, where Ours can achieve 700x speedup since Ours is a feed-forward method. Additionally, we provide a qualitative comparison on our website (please refer to "Response to Reviewer WQpa").
>
> | Method | Area | Time/Sample on a A100 (80G) |  Background   | Realism |
> | --- | :---:  | :---: |:---: | :---: |
> | TC4D   | 4D Gen. | 10h (3 per-scene optimized stages)  | &#10007;  | Medium |
> | Ours   | Video Gen.  | 0.8min (1 feed-forward stage)  | &#10003; | High |
>
> **2. Qualitative Feedback from Human Users**
>
> We conducted a questionnaire survey and collected 53 samples to form user preference comparisons. Each participant received a reward of 0.80 USD and spent approximately 5 minutes completing the questionnaire, which assessed four dimensions: (1) video quality, (2) trajectory accuracy, (3) entity diversity, and (4) background diversity. Below, we report the proportion of users who preferred our model over the baselines.
>
> | Method | MotionCtrl | Direct-a-Video | TC4D | Tora |
> | --- | :---:  | :---: |:---: | :---: |
> | **3DTrajMaster** | 47.2%  | 56.6% | 62.3% | 81.1% |
>
> **3. Real-world Application**
>
> We have expanded our evaluation to include diverse real-world scenarios and have provided generalizable videos (see "**W2 Generalizability**"). Additionally, we have explored various aspects from different domain experts and outlined our potential applications as follows.
>
> **1. Film**: Reproduce the character's classic moves. We can extract the human poses from a given video and apply them to different entities and backgrounds using the capabilities of our model (similar to that in "Response to Reviewer WQpa" on our website).
>
> **2. Autonomous Driving**: Simulate dangerous safety accidents, such as two cars colliding and a car hitting a person.
>
> **3. Embodied AI**: Generate a vast number of videos with diverse entity and trajectory inputs to train a general 4D pose estimator, especially for non-rigid objects.
>
> **4. Game**: Train a character ID, such as Black Myth Wukong, through LoRA, and then drive the character movement with different trajectories.

---

> ### Author Response · Authors · 2024-11-23
> **Awaiting Your Response (3 Days Left)**
>
> Dear Reviewer `WQpa`,
>
> Thank you once again for your time and effort in reviewing our work. With only `3 days` remaining, we kindly request your feedback on our response. If any part of our explanation is unclear, please let us know.
>
> We would greatly appreciate it if you could confirm whether your concerns have been addressed. If the issues are resolved, we would be grateful if you could consider reevaluating the work. Should you need any further clarification, we are happy to provide it promptly before the discussion deadline.
>
> Best,
>
> 3DTrajMaster Author(s) Team

---

> ### Author Response · Authors · 2024-11-25
> **Awaiting Your Response (Last 1 Day Left)**
>
> Dear Reviewer `WQpa`,
>
> Thank you once again for your time and effort in reviewing our work. With only `last 1 day` remaining, we kindly request your feedback on our response. If any part of our explanation is unclear, please let us know. As you mentioned, you would be reviewing other reviewers' feedback; we would appreciate it if you could also consider the responses we’ve provided as well as their corresponding replies.
>
> We would greatly appreciate it if you could confirm whether your concerns have been addressed. If the issues are resolved, we would be grateful if you could consider reevaluating the work. Should you need any further clarification, we are happy to provide it promptly before the discussion deadline.
>
> Best,
>
> 3DTrajMaster Author(s) Team

---

> ### Author Response · Authors · 2024-11-27
> **Awaiting Your Response (Approaching Discussion Extension DDL)**
>
> Dear Reviewer `WQpa`,
>
> Thank you once again for your time and effort in reviewing our work. **As the deadline for discussion extension DDL (Dec. 2) is approaching**, we kindly request your feedback on our response. If any part of our explanation is unclear, please let us know. As you mentioned, you would be reviewing other reviewers' feedback; we would appreciate it if you could also consider the responses we’ve provided as well as their corresponding replies.
>
> We would greatly appreciate it if you could confirm whether your concerns have been addressed. If the issues are resolved, we would be grateful if you could consider reevaluating the work. Should you need any further clarification, we are happy to provide it promptly before the discussion deadline.
>
> Best,
>
> 3DTrajMaster Author(s) Team

---

> ### Comment · Area_Chair_pbnj · 2024-12-01
>
> Dear Reviewer WQpa,
>
> As the authors have responded to your comments in the rebuttal, it would be helpful if you could provide your feedback so that the authors, peer reviewers, and ACs can know whether the concerns have been resolved or still remain.
>
> Thank you.
>
> AC

---

> > ### Author Response · Authors · 2024-12-04
> > **Thanks to AC for the Final Reminder (Yet Still No Response?)**
> >
> > Dear Reviewer `WQpa`,
> >
> > This is a gentle final reminder to review our response, as today is **the last opportunity** for our author to participate in any further discussions. As mentioned previously, you indicated, “I will also check other reviewers’ feedback,” and we truly hope that you can consider both our responses and the corresponding replies as part of your review process. Additionally, we believe it would be helpful to respond to the AC’s request, as a lack of communication could leave a negative impression in the final decision.
> >
> > We also apologize for any inconvenience caused to the other reviewers due to the repeated reminders sent to you. We understand you may have been busy and appreciate your time and attention to this matter.
> >
> > Wishing you all the best as approach the end of 2024.
> >
> > Warm regards,
> >
> > 3DTrajMaster Author(s) Team

---

### Official Review · Reviewer_zy5x · 2024-11-04

**Soundness:** 3
**Presentation:** 3
**Contribution:** 2
**Rating:** 6
**Confidence:** 3

**Summary:**

The paper proposes a 3D-trajectory-conditioned video generation method, fusing prior from pre-trained video diffusion models and from a proposed motion dataset.

**Strengths:**

The paper addresses the lack of 6-DoF controllability of existing video generation methods. The method is well-motivated and method designs are clearly explained. The advantage of 6-DoF control over 2D motion control is clearly demonstrated in experiments.

**Weaknesses:**

* The section on related works discusses prior methods on motion control and motion synthesis tasks, but could also include discussions on techniques for injecting controls to video foundation models, including ControlNet [1] and methods that allow 2D image editing by manipulation attention maps. In particular, ControlNet [1] is currently mentioned but not cited in the paper.
* The proposed dataset is restricted to human and animal categories, and locations remain to be in cities. Whether it's feasible to scale this method to generic object categories and generic scenes remains an open question.

[1] Lvmin Zhang, Anyi Rao, Maneesh Agrawala. Adding Conditional Control to Text-to-Image Diffusion Models.

**Questions:**

* Evaluation of multi-entity input sequence sets $N=2$, i.e., 2 entities, based on the qualitative examples. Is the method restricted to a small number of entities, and if so, does the restriction come from training data? If it's tied to training data distribution, it remains even more unclear if the method can potentially apply to generic settings where >>2 objects are moving, which is fairly common in dynamic scenes.

---

> ### Author Response · Authors · 2024-11-17
> **Response to Reviewer zy5x (Part 1/2)**
>
> Thanks for your constructive feedback and valuable time to review our work! We also kindly urge you to check our "Update on Warm-up Discussion, Anonymous Website, Paper Revision, and Dataset&Code Release" to catch up on our latest updates.
>
> Moreover, we address your concerns below.
>
> **(W1 Missing Related Work)**
>
> We fully agree with this insightful suggestion. Thank you! Below is the augmentation we have prepared.
>
> The methods for injecting control signals into video foundation models can be broadly classified into two categories:
>
> (1) Learning-based: The control signals are typically projected into latent embeddings via an extra encoder (e.g., learnable convolutional/linear/attention/LoRA layers, or frozen pre-trained feature encoder), which are then integrated into the base model architecture through concatenation, addition, or insertion. VideoComposer[1] employs a unified STC-encoder and CLIP model to feed multi-modal input conditions (textual, spatial, and temporal) into the base T2V model. MotionCtrl[2] introduces camera motion by fine-tuning specific layers of the base U-Net, and object motion via additional convolutional layers. CameraCtrl[3] enhances this approach by incorporating ControlNet[4]'s philosophy, using an attention-based pose encoder to fuse camera signals in the form of Plücker embeddings while keeping the base model frozen. Similarly, SparseCtrl[5] learns an add-on encoder to integrate control signals (RGB, sketch, depth) into the base model. Tora[6] employs a trajectory encoder and plug-and-play motion fuser to merge 2D trajectories with the base video model. MotionDirector[7] leverages spatial and temporal LoRA layers to learn desired motion patterns from reference videos.
>
> (2) Training-free: These methods modify attention layers or video latents to adjust control signals in a computationally efficient manner. However, training-free methods often suffer from poor generalization and require extensive trial-and-error. Direct-a-video[8] amplifies or suppresses attention in spatial cross-attention layers to inject box guidance, while FreeTraj[9] embeds target trajectories into the low-frequency components and redesigns reweighting strategies across attention layers. MOFT[10] extracts motion priors by removing content correlation and applying motion channel filtering, and then alters the sampling process using the reference MOFT.
>
> To enhance generalization, our 3DTrajMaster adopts a learning-based approach, specifically a plug-and-play object injector module that guides the base model to learn 3D motion patterns.
>
> [1] VideoComposer: Compositional Video Synthesis with Motion Controllability. ARXIV 2023
>
> [2] MotionCtrl: A Unified and Flexible Motion Controller for Video Generation. SIGGRAPH 2024
>
> [3] CameraCtrl: Enabling Camera Control for Video Diffusion Models. ARXIV 2024
>
> [4] Adding Conditional Control to Text-to-Image Diffusion Models. ICCV 2023
>
> [5] SparseCtrl: Adding Sparse Controls to Text-to-Video Diffusion Models. ECCV 2024
>
> [6] Tora: Trajectory-oriented Diffusion Transformer for Video Generation. ARXIV 2024
>
> [7] MotionDirector: Motion Customization of Text-to-Video Diffusion Models. ECCV 2024
>
> [8] Direct-a-Video: Customized Video Generation with User-Directed Camera Movement and Object Motion. SIGGRAPH 2024
>
> [9] FreeTraj: Tuning-Free Trajectory Control via Noise Guided Video Diffusion. ARXIV 2024
>
> [10] Video Diffusion Models are Training-free Motion Interpreter and Controller. NeurIPS 2024

---

> ### Author Response · Authors · 2024-11-17
> **Response to Reviewer zy5x (Part 2/2)**
>
> **(W2 Limited Dataset Scope)**
>
> **1. Scaling Up to Generic Scenes**
>
> We have extended the pure city dataset to more diverse 3D UE scenes such as desert, forest, Asian town, HDRIs (projected into 3D), as well as more complex 3D trajectory templates. Please see the dataset samples on our anonymous website. Below is a summary of the differences.
>
> | Dataset      | 3D Scenes | Samples     |  3D Trajectory Templates     |
> | --- | --- | :---: |:---: |
> | Rebuttal Version   | 9 (city, desert, forest, Asian town, 5 HDRIs)       | 54,000     | 96  |
> | Submission Version     | 1 (city)       | 12,000   | 48  |
>
> **2. Scaling Up to Generic Objects**
>
> First, we fully agree that scaling up to include more entities could further enhance the model generalizability. However, we think that the current set of 70 entities is already sufficient to generate a wide variety of diverse entities. Please refer to 'Control Entity Motion with Diverse Entities' and 'Control Multi-Entity Motion with Complex 3D Trajectories' on our website, where we can even generalize to control the motion of natural forces, such as fire and clouds, in 3D space.
>
> Moreover, we have updated the evaluation benchmark to include a broader range of entities. Below is a summary of the key differences, and for further details, please refer to the benchmark results on our website and Tables R4–R7.
>
> | Eval. Benchmark | Num. of Human Prompts  | Num. of Non-Human Prompts |  Num. of Location Prompts  |
> | --- | :---: | :---: |:---: |
> |  Rebuttal Version   | 20    | 52 (animal, car, robot)    |  32 |
> | Submission Version      | 20     | 40 (animal)  |  1 (city) |
>
>
> **(Q1 >>2 Multi-Entity Input)**
>
> **1. Clarification of Multi-Entity Evaluation**
>
> Actually, our multi-entity evaluation consists of 2 or 3 entities. We have provided the full benchmark results (raw data, ~200M) on our website.
>
> | Num. of All Entities | Num. of 1 Entity  | Num. of 2 Entities |  Num. of 3 Entities |
> | :---: | :---: | :---: |:---: |
> |  100 | 12  | 72 |  16 |
>
>
> **2. The Reason behind the Limited Entity Number**
>
> Currently, our method is limited to generating up to 3 entities, as outlined in the 'Limitation' section of the paper. This constraint is primarily due to the capabilities of the video foundation model rather than the training data. While it is relatively easy to generate >>2 entities of the same category (e.g., "a group of people/cars/animals") in the video, it becomes much more challenging to generate >>2 entities, each differing greatly from the others, through the text input as T5 text encoder tends to mix the textual features of different entities. Thus it becomes hard to associate specific trajectories with their corresponding text entities. Based on empirical studies with video foundation models, we chose to limit the number of entities to 3 in our work. Regarding data construction, it is easy to include more entities with their paired trajectories in our procedure UE platform pipeline. However, the key limitation is that the video foundation model struggles to generate such a diverse set of entities simultaneously. Furthermore, many prior works, such as Tora, MotionCtrl, and Direct-a-video also focus on a limited number of entities.
>
> If you have further questions or expected experiments, feel free to let us know. Besides, all reviewers are welcome to request extra video samples with specific entity prompts, trajectory templates, and locations, and we will generate those videos and make them available on the website.

---

> ### Author Response · Authors · 2024-11-23
> **Awaiting Your Response (3 Days Left)**
>
> Dear Reviewer `zy5x`,
>
> Thank you once again for your time and effort in reviewing our work. With only `3 days` remaining, we kindly request your feedback on our response. If any part of our explanation is unclear, please let us know.
>
> We would greatly appreciate it if you could confirm whether your concerns have been addressed. If the issues are resolved, we would be grateful if you could consider reevaluating the work. Should you need any further clarification, we are happy to provide it promptly before the discussion deadline.
>
> Best,
>
> 3DTrajMaster Author(s) Team

---

> > ### Comment · Reviewer_zy5x · 2024-11-24
> > **Thanks for Response**
> >
> > Thank you authors for the detailed response. The additional discussions on related works and additional results addressed my main concerns. I found the qualitative examples with multi-object-trajectory controls compelling. Qualitative results well support the claimed advantage of the method. Therefore I've updated my score to vote for acceptance.

---

> ### Author Response · Authors · 2024-11-24
> **Thanks for Your Recognization!**
>
> Dear Reviewer `zy5x`,
>
> Thank you once again for your thoughtful review and for providing a positive evaluation of our work in time. We greatly appreciate your valuable suggestions, which include:
>
> 1. The observation about missing related work indeed helps to strengthen the comprehensiveness of our paper and guide readers in understanding previous contributions in the field of video control.
>
> 2. The question regarding the scaling of the dataset has inspired us to demonstrate better the scalability of our proposed dataset construction approach and the potential generalizability of our model.
>
> 3. The inquiry regarding the infinite number of controlling entities has also prompted us to more clearly explain the underlying challenges and current limitations of video diffusion models.
>
> We firmly believe that our work will **encourage further research into fine-grained, 3D-aware motion control for video generation**.
>
> Best,
>
> 3DTrajMaster Author(s) Team

---

### Author Response · Authors · 2024-11-13
**Conclusion of Discussion Period, Paper Revision, and Dataset&Code Release (for AC's Meta Review Convenience)**

> **1. Conclusion of Discussion Period**

Our work is `the first to control 3D multi-entity motion in video generation`. In contrast to the previous 2D control signals, we propose to manipulate dynamics in 3D space, which can better express the 3D nature of object motions, with entity-specific 3D pose sequences (6 DoF) as additional inputs. We are pleased to see that the reviewers acknowledge the following aspects of our work.

1. **The first to control multi-entity motion with entity-specific 3D trajectories in video generation.** (Reviewer zy5x, WQpa, 2Hag)
2. **Clear advantage of 6-DoF control over 2D motion control** (Reviewer zy5x, WQpa, 2Hag, 95Hu)
3. **The method is reasonably designed, straight-forward and flexible** (Reviewer zy5x, WQpa, 2Hag, 95Hu)
4. **The contribution of 360-Motion Dataset** (Reviewer WQpa, 2Hag)
5. **Solid and convincing experiments** (Reviewer zy5x, 2Hag, 95Hu)
6. **Impressive generalizable results** (Reviewer 95Hu)

We make our anonymous website live at https://3dtrajmaster.github.io/. It includes all the videos in our paper, benchmark results, samples from the 360-Motion Dataset, and additional videos requested by the reviewers (we kindly invite all reviewers to **request extra video samples** with specific entity prompts, trajectory templates, and locations, and generate those videos and make them available on the anonymous website)

Regarding reviewers' concerns, we also have made efforts to address the reviewers' concerns in the relevant sections and reflect them in the Paper Revision.

_________________

> **2. Paper Revision**

**(1) More Diverse Training Dataset (Reviewer zy5x, WQpa, 2Hag)**: This is the primary limitation pointed out nearly by all reviewers. To address it, we extend the pure city dataset to more diverse 3D UE scenes such as desert, forest, Asian town, HDRIs (projected into 3D), as well as more complex 3D trajectory templates. Please see the dataset samples on our [anonymous website](https://3dtrajmaster.github.io/). Below is a summary of the differences.

| Dataset      | 3D Scenes | Samples     |  3D Trajectory Templates     |
| --- | --- | :---: |:---: |
| Rebuttal Version   | 9 (city, desert, forest, Asian town, 5 HDRIs)       | 54,000     | 96  |
| Submission Version     | 1 (city)       | 12,000   | 48  |

**(2) More Diverse Evaluation Benchmark (Reviewer zy5x, WQpa)**: Based on reviewers' doubts and Revision (1), we have expanded our evaluation to include a wider range of non-human and location prompts. Below is a summary of the differences. For more details, please refer to Tables R4-R7.

| Eval. Benchmark | Num. of Human Prompts  | Num. of Non-Human Prompts |  Num. of Location Prompts  |
| --- | :---: | :---: |:---: |
|  Rebuttal Version   | 20    | 52 (animal, car, robot)    |  32 |
| Submission Version      | 20     | 40 (animal)  |  1 (city) |

**(3) Further Revisions based on Reviewer 2Hag**:

1. Explanation of the"ControlNet-like architecture" (See ‘’W3-Minor Explanation of the 'ControlNet-like architecture' ")
2. Elaboration on Line 334 (see “W4-Minor Elaboration on Line 334”)
3. Elaboration on the “Limitation” section (see “2. Motion Type”)

**(4) Further Revisions based on Reviewer zy5x**:

1. Update missing related work (See "W1 Missing Related Work" and Sec. B in the paper)
2. Update "Clarification of the Limited Entity Number" in Sec. D of the paper
3. Clarify the multi-entity evaluation in Sec. 4.4

**(5) Further Revisions based on Reviewer WQpa**:

1. Update "Qualitative Feedback from Human Users" in Sec. F.2.3
2. Update "Additional Applications" in Sec. C

**(6) More SOTA baseline**: We additionally incorporate Tora [1], which is released after the ICLR submission DDL, in our comparison for more comprehensive results.

**(7) More Visually-Appealing Figures based on Revisions (1) and (2)**: We have redesigned the main paper figures to be more visually pleasing and adhere to our latest model requested by reviewers.

_________________

> **3. Dataset & Code Release**

Reviewer WQpa raises a question that we believe will be a concern for all reviewers. Below is our response.

**(1) Dataset Release**: Yes. We plan to release our dataset to support the research community once the paper is accepted, or possibly even earlier if the review scores are favorable. To demonstrate our commitment, we have provided a subset of smaller datasets on our anonymous website (including 5 scenes, and 480 videos).

**(2) Code Release**:  Yes. The release of the code based on our internal model will go through our internal license review process, but we commit to releasing a version based on the publicly available CogVideoX/Mochi 1 once the paper is accepted, or potentially even earlier if the review scores are favorable.

_________________

[1] Tora: Trajectory-oriented Diffusion Transformer for Video Generation. ARXIV, 2024

---

> ### Author Response · Authors · 2024-11-22
> **A Kind Reminder on Follow-up Discussion**
>
> Dear Reviewers,
>
> Thanks again for your constructive suggestions! As the discussion deadline (11.26) is approaching, we would like to send you a kind reminder. Some of you may be facing a tight deadline for submitting your CVPR paper. If that's the case, we hope everything goes smoothly and that you receive kind and constructive feedback from the reviewers as well.
>
> We were wondering whether you had the chance to look at our response and whether there is anything else you would like us to clarify. We sincerely hope that our response regarding your concerns will be taken into consideration. If not, please let us know and we remain open and would be more than glad to actively discuss them with you.
>
> Best,
>
> 3DTrajMaster author(s) team

---

### Meta-Review · Area_Chair_pbnj · 2024-12-16

**Metareview:**

This paper introduces a novel method for controlling 3D multi-entity motion in video generation. The proposed feed-forward approach allows for the manipulation of dynamics in 3D space using entity-specific 3D pose sequences (6 DOF) as additional inputs rather than relying on 2D control signals. Another significant contribution of this study is the creation of the 360-Motion Dataset, which the authors have committed to releasing along with the code.

The paper's main weaknesses include a lack of comparison with some state-of-the-art baselines, a need for a more diverse training dataset plus a diverse evaluation benchmark, and limitations on the number of entities for multi-entity input. The authors have addressed these concerns by providing extended data on different scenes, trajectory templates, and prompt types. The authors also conducted a user study and provided further explanations of weaknesses that they could not currently resolve.

The paper received ratings of (5, 6, 8, 8) from reviewers. Most expressed positive views regarding the paper's contributions, quality, and the significance of the experimental results. The area chair concurs with the reviewers' suggestions and recommends accepting the paper.

**Additional Comments On Reviewer Discussion:**

Most reviewers pointed out the diversity of the training data as the primary limitation. The authors addressed this issue by enhancing the dataset with more diverse 3D scenes and more complex 3D trajectory templates.

The authors adequately responded to a reviewer's request for discussions on techniques for injecting controls into video foundation models. They also provided a quantitative comparison with TC4D and a user study to compare with MotionCtrl, Direct-a-Video, TC4D, and Tora, although the proposed method did not show significant superiority.

Regarding the limitation of entity number, the authors provided a reasonable explanation as follows:
> Currently, our method is limited to generating up to 3 entities, as outlined in the 'Limitation' section of the paper. This constraint is primarily due to the capabilities of the video foundation model rather than the training data. While it is relatively easy to generate >>2 entities of the same category (e.g., "a group of people/cars/animals") in the video, it becomes much more challenging to generate >>2 entities, each differing greatly from the others, through the text input as T5 text encoder tends to mix the textual features of different entities. Thus it becomes hard to associate specific trajectories with their corresponding text entities. Based on empirical studies with video foundation models, we chose to limit the number of entities to 3 in our work.

---

### Decision · Program_Chairs · 2025-01-22

Accept (Poster)